# Conformal Alignment: Knowing When to Trust Foundation Models with Guarantees

**Yu Gui**[1]* **Ying Jin**[2]* **Zhimei Ren**[3]*

[1] Department of Statistics, University of Chicago
[2] Data Science Initiative, Harvard University
[3] Department of Statistics and Data Science, University of Pennsylvania
yugui@uchicago.edu, yjin@hcp.med.harvard.edu,
zren@wharton.upenn.edu

## Abstract

Before deploying outputs from foundation models in high-stakes tasks, it is imperative to ensure that they align with human values. For instance, in radiology report generation, reports generated by a vision-language model must align with human evaluations before their use in medical decision-making. This paper presents Conformal Alignment,[2] a general framework for identifying units whose outputs meet a user-specified alignment criterion. It is guaranteed that on average, a prescribed fraction of selected units indeed meet the alignment criterion, regardless of the foundation model or the data distribution. Given any pre-trained model and new units with model-generated outputs, Conformal Alignment leverages a set of reference data with ground-truth alignment status to train an alignment predictor. It then selects new units whose predicted alignment scores surpass a data-dependent threshold, certifying their corresponding outputs as trustworthy. Through applications to question answering and radiology report generation, we demonstrate that our method is able to accurately identify units with trustworthy outputs via lightweight training over a moderate amount of reference data. En route, we investigate the informativeness of various features in alignment prediction and combine them with standard models to construct the alignment predictor.

## 1 Introduction

Large-scale, pre-trained foundation models are remarkably powerful in generating relevant and informative outputs of various forms for diverse downstream tasks, marking a new era of artificial intelligence [34, 57, 47]. However, it has been recognized that they can be prone to factual errors [43], hallucinations [15, 51], and bias [4], among others. These issues raise prevalent societal concerns regarding the reliable use of foundation models in high-stakes scenarios [7, 6]. It remains a significant challenge to ensure that outputs from these highly complex models align with human values [36].

Towards uncertainty quantification of black-box models, conformal prediction [50, CP] offers a versatile, distribution-free solution. While CP has been applied to classification and regression tasks addressed by foundation models [25, 27, 46, 48, 54, 2], its use to ensure the alignment of general-form outputs remains largely unexplored. Recently, [36] propose to generate multiple outputs for one input until it is guaranteed that *at least* one output meets a specific alignment criterion. Such a guarantee, however, may not be immediately practical, as users must still determine (manually) which output(s) is aligned. [32] modify the machine-generated outputs to ensure factuality with

---

*Alphabetical ordering.
[2]The code is available at https://github.com/yugjerry/conformal-alignment.

high probability, which in turn may sacrifice model power by making responses vague.[3] As such, determining guarantees that are practically meaningful for downstream tasks and developing methods to achieve them remain an active research area.

This paper introduces a distinct type of guarantee: we aim to *certify* whether each model output is aligned or not, such that *those certified* are ensured to be *mostly correct*. This guarantee is directly relevant to downstream tasks, as it allows immediate and reliable use of the certified outputs without further modifications. On the other hand, we *abstain* from deploying model outputs for units (data points) that are not selected—intuitively, those are where the model lacks confidence (i.e., "knowing they do not know"), and they can be evaluated by human experts. Such a practice therefore provides an *automated* pipeline for the safe deployment of model-generated outputs.

**Present work.** We present Conformal Alignment (CA), a general framework that determines *when* to trust foundation model outputs with finite-sample, distribution-free guarantees. The framework is based on CP ideas, but unlike standard CP that searches the sampling space to construct a prediction set that *contains* a desired output, we leverage the quantified confidence as an instrument to *select* units whose already-generated outputs are trusted. Given any model and any alignment criterion, Conformal Alignment ensures that a prescribed proportion of the selected units' model outputs indeed meet the alignment criterion.[4] Concretely, suppose $m \in \mathbb{N}_+$ new units awaiting outputs. Given an error level $\alpha \in (0, 1)$, Conformal Alignment controls the false discovery rate (FDR) in selecting units with trustworthy outputs:

$$\text{FDR} = \mathbb{E}\left[ \frac{\sum_{i=1}^m \mathbb{1}\{i \text{ selected and not aligned}\}}{\sum_{i=1}^m \mathbb{1}\{i \text{ selected}\}} \right] \leq \alpha, \tag{1.1}$$

where the expectation is taken over the randomness of the data. FDR control offers an interpretable measure of the quality of selected deployable units. Our method builds upon the *Conformalized Selection* framework [18, 17], leveraging a holdout set of "high-quality" reference data to guide the selection with calibrated statistical confidence for trusted outputs. Like in CP, (1.1) holds in finite-sample as long as the holdout data are exchangeable with the new units.

Our method inherits CP's (1) **rigor**, providing distribution-free, finite-sample error control, and (2) **versatility**, applicable to any model and any alignment criterion. In addition, by selecting rather than modifying outputs, our approach preserves the (3) **informativeness** of the original outputs, and remains (4) **lightweight**, e.g., avoiding the need to retrain large models.

**Demonstration of workflow.** We illustrate the pipeline of Conformal Alignment via a use case in radiology report generation (detailed in Section 5) in Figure 1: each prompt corresponds to an X-ray scan, for which a vision-language model generates a report. Our framework identifies a subset of generated reports that are reliable in the sense that at least, say 99%, of the selected reports are *guaranteed* to be aligned if they were to be compared with a human expert report.

The workflow begins with (test) prompts $\mathcal{D}_{\text{test}}$, corresponding to X-ray scans $X \in \mathcal{X}$. A foundation model $f \colon \mathcal{X} \mapsto \mathcal{Y}$ generates a report $f(X) \in \mathcal{Y}$ for each scan. However, only high-quality reports should be handed over to doctors for clinical decisions. To achieve this, we leverage a set of X-ray scans that are independent of the model's training process, each with human expert reports available. These scans are randomly split into a training set $\mathcal{D}_{\text{tr}}$ and a calibration set $\mathcal{D}_{\text{cal}}$. Conformal Alignment trains a predictor on $\mathcal{D}_{\text{tr}}$ for predicting the alignment score that assess how likely a generated output aligns. Finally, new reports are selected if their predicted alignment scores exceed a *data-driven* threshold, which is delicately set with $\mathcal{D}_{\text{cal}}$ such that the FDR is strictly controlled at the desired level.

**Practical relevance.** We apply Conformal Alignment to two use cases—question answering and radiology report generation—to demonstrate its efficiency. With quite lightweight training of the alignment predictor (for example, training a tree-based model would suffice), we are able to accurately identify trustworthy outputs and select as many of them as possible. En route, we exhaustively investigate practical aspects such as the efficacy of various predictors in these applications and the amount of "high-quality" data needed for accurate selection to guide practical deployment.

---

[3]See Appendix D.7 for more detailed discussion on the types of guarantees.

[4]In this work, we use "alignment" to refer to a desired property of an output that may vary with the context.

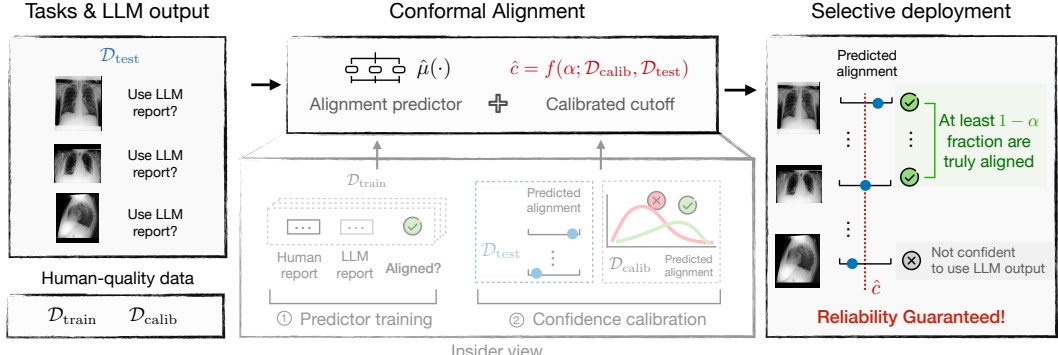

Figure 1: Pipeline of Conformal Alignment instantiated in the radiology report generation example.

## 2 Problem setup

Suppose we have a pre-trained foundation model $f : \mathcal{X} \mapsto \mathcal{Y}$ that maps a prompt to an output. Throughout, we view the model as given and fixed. Assume access to a set of holdout units $\mathcal{D} = (X_i, E_i)_{i=1}^{n}$, where $X_i \in \mathcal{X}$ is the input prompt and $E_i \in \mathcal{E}$ is any reference that may be used in judging alignment. An *alignment function* $\mathcal{A} : \mathcal{Y} \times \mathcal{E} \mapsto \mathbb{R}$ takes as input the generated output $f(X)$ and the reference $E$, and outputs an (true) alignment score $A = \mathcal{A}(f(X), E)$ [36]. In the radiology report generation example, $A$ might be the similarity score between the machine-generated report $f(X)$ and a human expert report $E$. We shall detail the choice of $\mathcal{A}$ in each use case. Note that for a unit without reference information, the alignment score of its generated output is unknown, and we do not seek to evaluate it, which often requires expert annotation or human judgment.

For a test set $\mathcal{D}_{\text{test}} = \{X_{n+j}\}_{j=1}^{m}$, we aim to select a subset $\mathcal{S} \subseteq [m] := \{1, \ldots, m\}$ such that most of their (unobserved) alignment scores exceed a pre-fixed threshold $c \in \mathbb{R}$. The error is measured by

$$\text{FDR} = \mathbb{E}\left[\frac{\sum_{j \in [m]} \mathbb{1}\{A_{n+j} \le c, j \in \mathcal{S}\}}{\max(|\mathcal{S}|, 1)}\right],$$

where $|\cdot|$ denotes the cardinality of a set. In particular, we aim to enforce that FDR $\le \alpha$ for a pre-specified level $\alpha \in (0, 1)$. The FDR measures the averaged proportion of *selected units* that are not aligned, thereby directly quantifying the "cost" of deploying outputs in $\mathcal{S}$. Such a measure (under different terminologies) appears to be considered empirically in selective prediction, although a rigorous control has been lacking therein (see e.g., [52, 49, 37]). Besides FDR control, it is also desirable that as many units can be safely deployed, which translates into maximizing the power:

$$\text{Power} = \mathbb{E}\left[\frac{\sum_{j \in [m]} \mathbb{1}\{A_{n+j} > c, j \in \mathcal{S}\}}{\max(\sum_{j \in [m]} \mathbb{1}\{A_{n+j} > c\}, 1)}\right]. \tag{2.1}$$

It should be emphasized that our framework prioritizes FDR control over power, in that we strictly enforce the former while optimizing the latter under the constraints. This is related to but slightly differs from the *selection with controlled risk* setting in selective prediction [11, 12]; see Section B.1 for more details. Such a setup is motivated by scenarios such as medical decision-making and knowledge access, where the cost of committing a type-I error (i.e., selecting a non-aligned unit) can be grave, and should be of primary concern. In addition, we define power as the proportion among truly aligned units, instead of the number of selections—indeed, if a foundation model is not powerful in generating aligned outputs, to begin with, it is natural that only a few of the outputs shall be deployed.

### 2.1 Related works

**Uncertainty quantification and conformal prediction for foundation models.** Conformal prediction [50, 26, 41] is a distribution-free framework for uncertainty quantification of generic prediction algorithms. This work adds to the growing literature that applies/extends CP to foundation models,

which varies in definitions of uncertainty and alignment, types of guarantees, rules for decision-making, etc. Below, we summarize representative ideas and contrast them with the present work.

A line of work applies CP to construct prediction sets that cover the outcome for a *single* test unit in classification or regression problems addressed by language models [25, 27, 46, 48, 54]. While desired, their reliability *in downstream tasks* is only empirically evaluated: we note that coverage of prediction sets *does not* carry over to the selected units [18, 19]. In contrast, this work applies to outputs of general forms, considers multiple units simultaneously, and offers guarantees relevant to downstream tasks. In particular, our methodology is built upon the series of papers in *conformalized selection* [18, 17]. While they aim to find large values of outcomes, we motivate and tailor the framework for the alignment of foundation models (a more detailed discussion is in Appendix B.2).

Two works [36, 32] addressing the alignment of foundation models with CP ideas have been briefly discussed in Section 1. Our setup draws ideas from [36], but the guarantees we provide differ from both. In particular, our method may be better suited to situations where it is desirable to avoid modifying the outputs or enlarging candidate sets which requires retraining the models.

More generally, [24, 29, 16] investigate the metric of uncertainty for natural language outputs. This is orthogonal to the present work, as our method is compatible with any alignment criterion. Nevertheless, our work adds to the discussion on desirable uncertainty quantification guarantees.

**Selective prediction.**   Our framework is closely related to the idea of selective prediction, where one is allowed either to *predict* or to *abstain* from making a prediction [9, 11, 12, 33, 1]. In the context of large language models, [23, 53, 40, 55] have similarly considered the goal of teaching the model not to predict when the model is uncertain about its output, but theoretical guarantees on the alignment of the selected outputs are lacking. In contrast, this work rigorously formalizes the guarantee and provides an end-to-end framework that achieves it under a mild exchangeability condition. As mentioned earlier, the strict control of FDR is closely connected to the "selection with controlled risk" setting in selective prediction [11, 12], which aims to control a slightly different error measure; we provide a detailed comparison of the error measures and methods in Appendix B.1.

**Alignment of foundation models.**   There is a rapidly growing literature on the alignment of foundation models, wherein topics include alignment evaluation [30, 42], prompt design [22], training with human feedback [3, 35, 38], among others [8]. We rely on commonly-used alignment criteria in our applications [44, 29, 24]. However, we achieve strict error control via post-hoc adjustment/selection, instead of deriving training strategies to improve certain alignment metrics of the model.

## 3   Conformal Alignment

Given a set of data $\mathcal{D}$ with reference information, our procedure starts by splitting $\mathcal{D}$ into two disjoint sets: the training set $\mathcal{D}_{\mathrm{tr}}$ and the calibration set $\mathcal{D}_{\mathrm{cal}}$. With a slight abuse of notation, we shall also use $\mathcal{D}_{\mathrm{tr}}$ and $\mathcal{D}_{\mathrm{cal}}$ to denote the indices of the units in the sets when the context is clear. We then fit a model $g : \mathcal{X} \to \mathbb{R}$ on $\mathcal{D}_{\mathrm{tr}}$ to predict the alignment score based on $X$ (which may also involve information from $f$). Next, we generate model outputs $f(X_i)$ and compute the predicted alignment scores $\widehat{A}_i = g(X_i)$ for every $i \in [n + m]$. We shall use $(A_i, \widehat{A}_i)_{i \in \mathcal{D}_{\mathrm{cal}}}$ to determine the selection set.

Following the framework of *conformalized selection* [18, 17], we gather statistical evidence for trustworthy outputs via hypothesis testing, where the null hypothesis for $j \in [m]$ is

$$H_j : \ A_{n+j} \leq c. \tag{3.1}$$

Rejecting $H_j$ then reflects evidence that the (true) alignment score of unit $j$ is above the threshold $c$, and therefore the generated output $f(X_{n+j})$ is aligned. Under this framework, the task of selecting aligned units boils down to simultaneously testing the $m$ hypotheses specified in (3.1). For this purpose, we construct the *conformal p-value*: for any $j \in [m]$,

$$p_j = \frac{1 + \sum_{i \in \mathcal{D}_{\mathrm{cal}}} \mathbb{1}\{A_i \leq c, \widehat{A}_i \geq \widehat{A}_{n+j}\}}{|\mathcal{D}_{\mathrm{cal}}| + 1}. \tag{3.2}$$

It can be shown that when the test unit is exchangeable with the calibration units, the p-value defined above is valid in the sense that $\mathbb{P}(p_j \leq t, A_{n+j} \leq c) \leq t$ for any $t \in (0, 1)$ [18]. Intuitively, when a generated output is likely to be aligned, we expect $\widehat{A}_{n+j}$ to have a large magnitude, and $p_j$ to be

small. As a result, we reject the null hypothesis—that is, declaring a sufficiently large alignment score—for a small $p_j$, where the threshold for p-values is determined by the Benjamini-Hochberg (BH) procedure [5]. Concretely, let $p_{(1)} \leq \ldots p_{(m)}$ denote the ordered statistics of the p-values; the rejection set of BH applied to the conformal p-values is $\mathcal{S} = \{j \in [m] : p_j \leq \alpha k^*/m\}$, where

$$k^* = \max\left\{k \in [m] : p_{(k)} \leq \frac{\alpha k}{m}\right\},$$

with the convention that $\max \varnothing = 0$. We describe the complete procedure in Algorithm 1, and establish its validity in Theorem 3.1. For notational simplicity, we denote $Z_i = (X_i, E_i)$ for all $i \in [n + m]$ (note that $E_i$ is not observable for $i > n$).

---

**Algorithm 1** Conformal Alignment

**Input:** Pre-trained foundation model $f$; alignment score function $\mathcal{A}$; reference dataset $\mathcal{D} = (X_i, E_i)_{i=1}^n$; test dataset $\mathcal{D}_{\text{test}} = (X_{n+j})_{j=1}^m$; algorithm for fitting alignment predictor $\mathcal{G}$; alignment level $c$; target FDR level $\alpha$.
1: Compute the alignment score $A_i = \mathcal{A}(f(X_i), E_i), \forall i \in \mathcal{D}$.
2: Randomly split $\mathcal{D}$ into two disjoint sets: the training set $\mathcal{D}_{\text{tr}}$ and the calibration set $\mathcal{D}_{\text{cal}}$.
3: Fit the alignment score predictor with $\mathcal{D}_{\text{tr}}$: $g \leftarrow \mathcal{G}(\mathcal{D}_{\text{tr}})$.
4: Compute the predicted alignment score: $\widehat{A}_i \leftarrow g(X_i), \forall i \in \mathcal{D}_{\text{cal}} \cup \mathcal{D}_{\text{test}}$.
5: **for** $j \in [m]$ **do**
6:    Compute the conformal p-values $p_j$ according to Equation (3.2).
7: **end for**
8: Apply BH to the conformal p-values: $\mathcal{S} \leftarrow \text{BH}(p_1 \ldots, p_m)$.
**Output:** The selected units $\mathcal{S}$.

---

**Theorem 3.1.** Suppose that for any $j \in [m]$, $\{Z_{n+j}\} \cup \{Z_i\}_{i \in \mathcal{D}_{\text{cal}}}$ are exchangeable conditional on $\{Z_{n+\ell}\}_{\ell \neq j}$, i.e., for any permutation $\pi$ of $\{1, \ldots, n, n+j\}$ and any $\{z_1, \ldots, z_n, z_{n+j}\}$, it holds that

$$\mathbb{P}\big(Z_1 = z_1, \ldots, Z_n = z_n, Z_{n+j} = z_{n+j} \,|\, \{Z_{n+\ell}\}_{\ell \neq j}\big)$$
$$= \mathbb{P}\big(Z_1 = z_{\pi(1)}, \ldots, Z_n = z_{\pi(n)}, Z_{n+j} = z_{\pi(n+j)} \,|\, \{Z_{n+\ell}\}_{\ell \neq j}\big).$$

Suppose the predicted alignment score $\{\widehat{A}_i\}_{i \in \mathcal{D}_{\text{cal}} \cup \mathcal{D}_{\text{test}}}$ have no ties almost surely and $\sup_x |g(x)| \leq \bar{M}$ for some $\bar{M} > 0$. Then for any $\alpha \in (0, 1)$, the output $\mathcal{S}$ from Algorithm 1 satisfies FDR$\leq \alpha$.

The proof of Theorem 3.1 is adapted from [18], and we include it in Appendix A for completeness.

**Remark 3.2.** The exchangeability condition required in Theorem 3.1 is satisfied when the samples in $\mathcal{D}_{\text{cal}} \cup \mathcal{D}_{\text{test}}$ are i.i.d., or when $\mathcal{D}_{\text{cal}}$ is drawn from a set of samples without replacement, or when they are nodes on graphs in the transductive setting [14]. The assumption of no ties is without loss of generality, since one can always add a small random noise to break the ties.

The following proposition characterizes the power (2.1), as well as the fraction of selected units that can be deployed with confidence, when the samples are i.i.d. The proof is in Appendix A.2. A finite-sample power analysis based on concentration inequalities is in Appendix A.3.

**Proposition 3.3.** Under the same condition of Theorem 3.1, assume further that $(X_i, E_i)_{i \in [n+m]}$ are i.i.d. Define $H(t) = \mathbb{P}(A \leq c, g(X) \geq t)$ and $t(\alpha) = \sup\{t : \frac{t}{\mathbb{P}(H(g(X)) \leq t)} \leq \alpha\}$. Suppose that there exists a sufficiently small $\varepsilon > 0$ such that $\frac{t}{\mathbb{P}(H(g(X)) \leq t)} < \alpha$ for $t \in [t(\alpha) - \varepsilon, t(\alpha)]$, then

$$\lim_{|\mathcal{D}_{\text{cal}}|, m \to \infty} \text{Power} = \mathbb{P}(H(g(X)) \leq t(\alpha) \,|\, A > c).$$

$$\lim_{|\mathcal{D}_{\text{cal}}|, m \to \infty} \frac{1}{m} \sum_{j=1}^m \mathbb{1}\{j \in \mathcal{S}, A_{n+j} > c\} = \mathbb{P}(H(g(X)) \leq t(\alpha), A > c).$$

Figure 2 visualizes the asymptotic cutoff on $g(X)$ beyond which will be selected by our method: it is the smallest value such that the red area on the right is less than $\alpha$-proportion of red and blue areas combined. The area of the blue part on the right of the cutoff is thus the asymptotic fraction of selected units, whose proportion among the entire blue area is the asymptotic power. Intuitively,

given a model (i.e., fixing areas for the red and blue), the power of Conformal Alignment depends on how well $g$ discerns those $A > c$ against those $A \leq c$. Therefore, it is important to identify features that informs the alignment of the outputs; we will elaborate on this point and deliver practical recommendations in our experiments in Sections 4 and 5. In addition, the fraction of deployable units (blue) additionally depends on the model power, i.e., the fraction of units whose outputs are indeed aligned. These two factors both affect the fraction of units selected by our method (the blue area on the right of the cutoff), which will also be demonstrated in the experiments.

## 4  Experiments for question answering (QA)

In this section, we implement and evaluate Conformal Alignment in question-answering tasks, where we consider a conversational question answering dataset **TriviaQA** [21] and a closed-book reading comprehension dataset **CoQA** [39]. For each QA dataset, we use language models **OPT-13B** [57] and **LLaMA-2-13B-chat** [47] without finetuning to generate an answer $f(X_i)$ via top-p sampling for each input $X_i$ following the default configuration. The alignment score $A_i$ measures how well the generated answer matches the true reference answers provided in both datasets. Following [24, 29], we let $A_i \in \{0, 1\}$ indicate whether the `rouge-L` score [28] between the LLM-generated answer and reference answer is no less than 0.3, and set $c = 0$. See Appendix C.1 for more details.

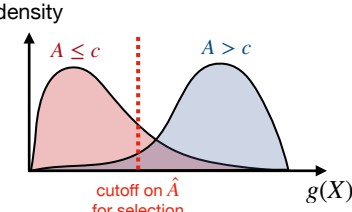

Figure 2: Visualization of asymptotic selection rule (red dashed line), with density curves of $g(X)$ for $A \leq c$ (red) and $A > c$ (blue).

**Dataset partition.**   Recall $\mathcal{D}$ is the reference dataset and $\mathcal{D}_{\text{test}}$ is the test set we select from. We fix $|\mathcal{D}_{\text{test}}| = 500$ and split $\mathcal{D}$ as follows. Fixing $\gamma_1, \gamma_2 \in (0, 1)$, $\gamma_1 + \gamma_2 < 1$, we randomly sample $\lfloor (\gamma_1 + \gamma_2) \cdot |\mathcal{D}| \rfloor$ instances without replacement from $\mathcal{D}$ as $\mathcal{D}_{\text{tr}}$ for training the alignment predictor $g$. Within $\mathcal{D}_{\text{tr}}$, a random subset of size $\lfloor \gamma_1 \cdot |\mathcal{D}| \rfloor$ is reserved for hyperparameter tuning in computing certain features to be introduced later, while the others are used in fitting $g$ given the features. We then set the calibration set $\mathcal{D}_{\text{cal}} = \mathcal{D} \backslash \mathcal{D}_{\text{tr}}$. For the results presented in this section, $\gamma_1 = 0.2$, $\gamma_2 = 0.5$. Additional ablation studies for the choice of $(\gamma_1, \gamma_2)$ are summarized in Section 4.1.

**Alignment score predictor.**   With the training set $\mathcal{D}_{\text{tr}}$, according to Algorithm 1, our goal is to train an alignment score predictor $g$ that maps $X_i$ to $\widehat{A}_i$—the estimated probability for $A_i > c$—based on available information such as the input, model parameters, and outputs. To this end, we train a classifier with binary label $A_i$ and the following features (as preparation, we additionally randomly generate 19 answers for each input, leading to $M = 20$ answers in total for each input):

- *Self-evaluation likelihood* (`Self_Eval`). Following [22, 29], we ask the language model itself to evaluate the correctness of its answer and use the likelihood, `P(true)`, as a measure of confidence.

- *Input uncertainty scores* (`Lexical_Sim`, `Num_Sets`, `SE`). Following [24], we compute a set of features that measure the uncertainty of each LLM input through similarity among the $M = 20$ answers. The features include lexical similarity (`Lexical_Sim`), i.e., the `rouge-L` similarity among the answers. In addition, we use a natural language inference (NLI) classifier to categorize the $M$ answers into semantic groups, and compute the number of semantic sets (`Num_Sets`) and semantic entropy (`SE`). Following [24, 29], we use an off-the-shelf DeBERTa-large model [13] as the NLI predictor.

- *Output confidence scores* (`EigV(J/E/C)`, `Deg(J/E/C)`, `Ecc(J/E/C)`). We also follow [29] to compute features that measure the so-called output confidence: with $M$ generations, we compute the eigenvalues of the graph Laplacian (`EigV`), the pairwise distance of generations based on the degree matrix (`Deg`), and the Eccentricity (`Ecc`) which incorporates the embedding information of each generation. Note that each quantity is associated with a similarity measure; we follow the notations in [29] and use the suffix J/E/C to differentiate similarities based on the Jaccard metric, NLI prediction for the entailment class, and NLI prediction for the contradiction class, respectively.

We use all above features to train the alignment predictor with standard ML models including logistic regression, random forests, and XGBoost. Existing works often use individual features in calibrating model confidence [22, 29, 24]; in this regard, we evaluate these features by using each individual feature to train the predictor $g$ and reporting the resulting power, which delivers additional insights upon their informativeness in predicting model alignment.

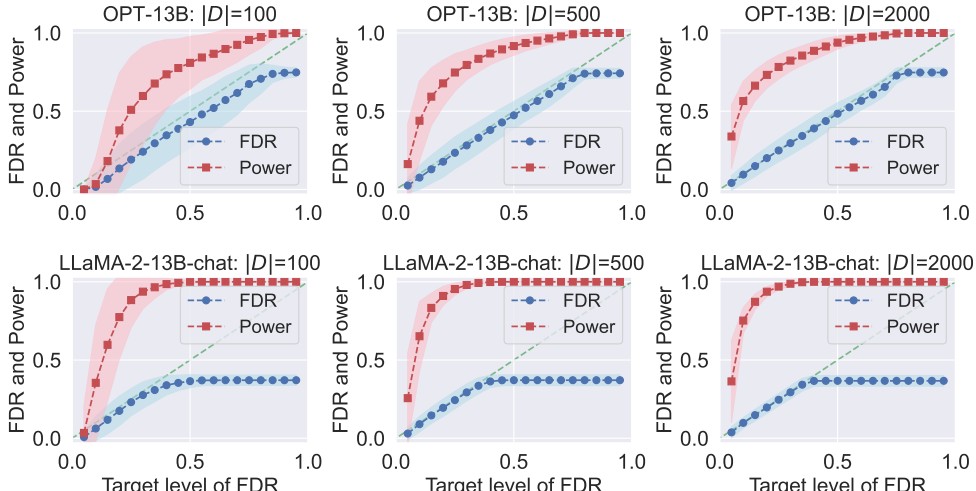

Figure 3: Realized FDR (blue) and power (red) for conformal alignment applied to the TriviaQA dataset (with $\gamma_1 = 0.2, \gamma_2 = 0.5$) at various FDR target levels. The top row corresponds to the results from OPT-13B and the bottom row to those from LLaMa-2-13B-chat; each column corresponds to a value of $|\mathcal{D}|$. Shading represents the area between one standard deviation above and below the mean.

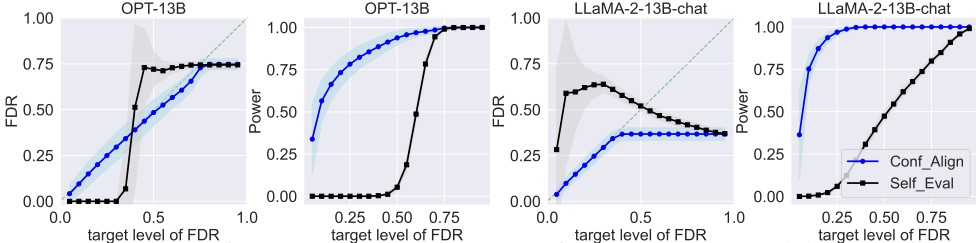

Figure 4: Comparison of FDR and power between Conformal Alignment and the heuristic baseline where we select units by thresholding self-evaluation scores `Self_Eval` with a cutoff at $1-$the target FDR level. The comparison is conducted on the TriviaQA dataset.

## 4.1 Experiment results

This section reports the results on the TriviaQA dataset, where a subset of $8000$ questions are considered, with the alignment predictor trained via *logistics regression*; additional results when the predictor is trained via random forest and XGBoost are in Appendix D.1. Parallel results on the CoQA dataset with 7700 questions can be found in Appendix D.2. We also provide QA examples in Table 1 in Appendix D.5 to illustrate the performance of our method.

**Conformal Alignment strictly controls the FDR while heuristic baseline does not.** Figure 3 shows the realized FDR and power for Conformal Alignment at target FDR levels $\alpha \in \{0.05, 0.1, \dots, 0.95\}$, averaged over $500$ independent experiments. The FDR curves demonstrate the finite-sample error control. It is worth noting that the FDR is tightly controlled; it becomes a constant for sufficiently large $\alpha$ under which all units can be selected without violating the FDR.

To demonstrate the importance of a statistically rigorous treatment, we compare Conformal Alignment with a heuristic baseline, where one asks the language model to evaluate its confidence (i.e., the `Self_Eval` feature above), and threshold the obtained likelihood $\{q_j^{\text{self}}\}$ with a cutoff of $1 - \alpha$ as if it were the "probability" of alignment, i.e., $\mathcal{S}_{\text{baseline}} = \{j \in [m] : q_j^{\text{self}} \geq 1 - \alpha\}$. The resulting FDR and power for the TriviaQA dataset are in Figure 4. The baseline fails to control FDR, showing that the self-evaluation score is over-confident. In addition, it is also less powerful than our procedure despite the higher error rate, implying that the self evaluation is less informative for predicting alignment. Comparing LLaMA-2-13B-chat and OPT-13B, we also observe that more powerful models may tend to be more (over-)confident in their output.

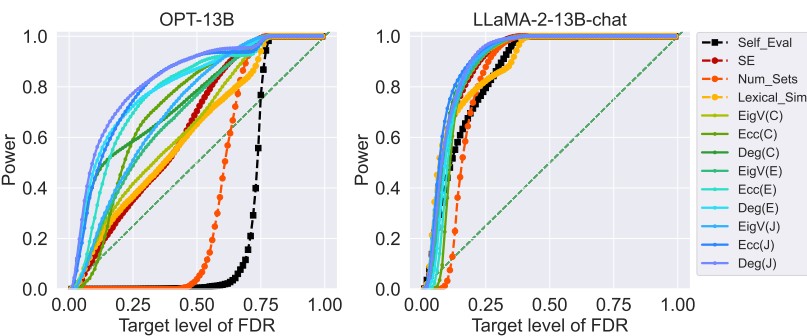

Figure 5: Power versus target FDR levels for TriviaQA dataset when the alignment predictor is trained with logistic regression over individual features with $|\mathcal{D}| = 2000$, $\gamma_1 = 0.2$, $\gamma_2 = 0.5$, averaged over 500 independent experiments. Note that the FDR is always controlled though not depicted.

**More powerful model enables more powerful selection.** The performance of Conformal Alignment also varies with the language models. We observe that more powerful LLMs can generate answers with higher accuracy which further leads to stronger signals in selection. For OPT-13B, the largest value of FDR is 0.5; we have nearly no discoveries and the power is close to zero when $\alpha = 0.05$. In comparison, with LLaMA-2-13B-chat, FDR never exceeds 0.25, and the power is as high as 0.50 even at an FDR level of $\alpha = 0.05$. As such, one may also use the FDR/power curve from Conformal Alignment as a measure to compare LLMs' performance.

**What feature is informative for alignment?** In addition to the capability of LLMs, the features used to train $g$ also play a vital role in the power of selection. To compare the informativeness of the aforementioned features for the alignment of outputs, we use one score at a time as the feature in training $g$ and show Power/FDR curves in Figure 5. When using the OPT-13B model, `Self_Eval` and `Num_Sets`s are the least powerful, while `Deg(J)` and `Ecc(J)` are the most powerful in predicting alignment. Such a comparison is similar for LLaMA-2-13B-chat. We defer the FDR curves to Figure 14 in Appendix D.3 which remain strictly controlled. We include a more detailed analysis on feature importance via Shapley value in Appendix D.6.

**How many high-quality samples are needed?** The size of high-quality samples $|\mathcal{D}|$ needed is important for applying Conformal Alignment. In practice, it is desired if a handful of data suffices to accurately identify aligned outputs. To study the effect of reference sample size $|\mathcal{D}|$ on the performance, in Figure 3, we vary $|\mathcal{D}|$ in $\{100, 500, 2000\}$ with fixed $\gamma_1 = 0.2$ and $\gamma_2 = 0.5$. In general, *a few hundred* high-quality samples suffice to achieve relatively high power, showing that Conformal Alignment does not require too expensive extra labeling. More specifically, we observe that when $|\mathcal{D}|$ increases, there is a slight improvement in power, especially when $\alpha$ is small; for large values of $\alpha$, the power is robust to $|\mathcal{D}|$. In addition, a larger labeled sample size reduces the variance in both FDR and Power and improves the stability in selection.

**Ablation study.** In Appendix D.4, we present ablation studies for the choice of $\gamma_1$ and $\gamma_2$ in data splitting. In practice, larger values of $\gamma_1$ and $\gamma_2$ allow more samples for training the alignment predictor but decrease the resolution of the conformal p-values. With fixed $|\mathcal{D}|$ and $\gamma_2$, we observe that the performance of our method is not sensitive to the choice of $\gamma_1$; in particular, a small value of $\gamma_1$ (e.g., 0.2) is sufficient for powerful selection. When fixing $|\mathcal{D}|$ and $\gamma_1 = 0.2$, we suggest setting $\gamma_2 \in [0.3, 0.5]$ to balance the sample sizes for both the predictor training and the BH procedure.

## 5   Experiments for chest X-ray report generation

In this section, we apply Conformal Alignment to chest X-ray report (CXR) generation. Following the pipeline in Figure 1, we apply our method to (a subset of) the MIMIC-CXR dataset [20]. In practice, imagine a healthcare institute employs a foundation model to automatically generate radiology reports based on chest radiographs. Conformal Alignment can be used to determine which reports to trust and refer to doctors for medical decisions, and the rigorous guarantees it offers imply that $1 - \alpha$ fraction of the approved reports indeed match the results if they were to be read by a human expert.

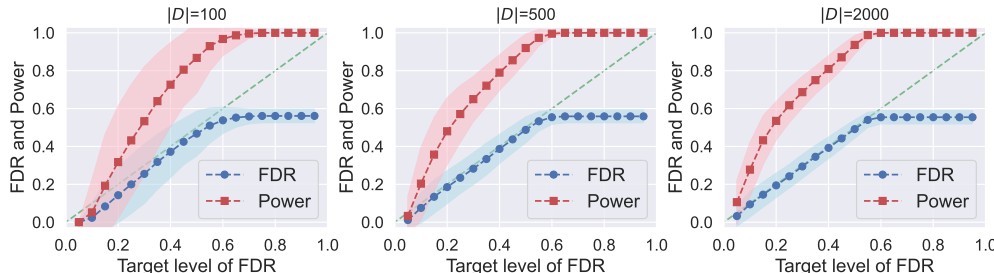

Figure 6: Realized FDR (blue) and power (blue) for Conformal Alignment applied to CXR report generation with alignment predictor from logistic regression. The results are averaged over 500 runs with $\gamma_1 = 0.2, \gamma_2 = 0.5$. Each subplot corresponds to one value of $|\mathcal{D}|$.

We obtain the base foundation model $f$ by fine-tuning an encoder-decoder model (pre-trained Vision-Transformer and GPT2) using a separate training set. For a generated report $f(X_i)$ with a reference report written by a radiologist, the alignment score $A_i$ is defined as follows: we use CheXbert [44] to convert both reports to two 14-dimensional vectors of binary labels with each coordinate as the indicator of positive observations; we then define $A_i \in \{0, 1\}$ as the indicator of whether there are at least 12 matched labels, and set $c = 0$. We employ the same set of features and the same procedure in Section 4, except for `Self_Eval` which is not available from the current model. More details about the experimental setup are in Appendix C.2.

## 5.1 Experiment results

We summarize the main findings from our experiments below; additional results with random forest and XGBoost predictors, as well as ablation studies for the choice of hyperparameters, are deferred to Appendix E. To illustrate the role of the alignment score predictor, we also present examples of CXR images with reference and generated reports in Table 2 of Appendix E.4.

**Tight FDR control.** Figure 6 presents the FDR and power curves for Conformal Alignment where the alignment predictor is trained with all features using logistic regression, averaged over 500 independent runs of the procedure. Again, we observe that the FDR is tightly controlled at the target level; it remains constant for sufficiently large $\alpha$ since by then all units can be selected without violating the FDR. We also observe satisfactory power, although it is lower than that is the QA tasks since CXR generation can be a more challenging task, and the model in use is fine-tuned with a limited number of training samples. We shall expect even higher power for better tuned models.

**Hundreds of reference data suffice.** Comparing subplots in Figure 6, as the size of the reference data $\mathcal{D}$ increases, we observe a slight improvement in power. However, we again find that $|\mathcal{D}| = 500$ already suffices for powerful deployment of Conformal Alignment. In practice, we expect this sample size also suffices for more powerful foundation models that are more carefully designed for this specific task.

**Informativeness of individual features.** To study the informativeness of different features for alignment, we use one feature at a time to train the logistic regression classifier and compare the resulting power versus target FDR levels in Figure 7 from Conformal Alignment (curves showing FDR control are in Figure 23 of Appendix E.2). In the CXR task, features based on Jaccard similarity are still the most powerful while those based on NLI

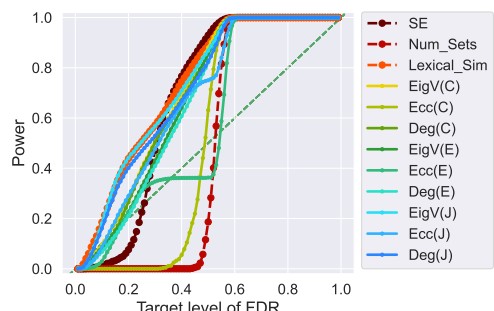

Figure 7: Power at each FDR level for CXR dataset when the alignment predictor is trained with logistic regression over each individual feature, fixing $|\mathcal{D}| = 2000, \gamma_1 = 0.2, \gamma_2 = 0.5$.

prediction exhibit slightly lower power. More complicated texts and more tokens in CXR reports could be potential reasons for the relatively poor performance of NLI prediction.

# 6 Discussion and limitations

We have presented Conformal Alignment, a principled framework for ensuring the alignment of foundation model outputs with provable, distribution-free FDR control. Compared with standard conformal prediction, we provide a solution that may be more relevant to downstream tasks than existing ideas. We demonstrate that our framework is flexible, lightweight, and effective through concrete applications to question answering and radiology report generation tasks, where we also provide practical recommendations on sample size, data splitting, and feature engineering.

The choice of alignment score in different applications remains an open problem. For example, in the radiology report generation example, as pointed out by [56], there can be a discrepancy between the score given by CheXbert and that given by human evaluation; as such, it will be interesting to use our framework with actual human evaluation labels. More generally, interesting future directions include designing implementation tailored for other appropriate applications, tackling parallel or sequential outputs, improving power in data-scarce scenarios (e.g., with better base models and more informative features), extending to multi-task and in-context learning settings, and generalizing our framework to control other error notions beyond FDR that may be meaningful for downstream decisions.

## Acknowledgments and Disclosure of Funding

Z.R. is supported by NSF grant DMS-2413135.

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

## Broader impact

It is crucial to ensure that LLM outputs are aligned with human values, especially in high-stakes scenarios (e.g. medical decision-making). This paper provides a generic framework that identifies reliable LLM outputs and controls the misalignment error rate in a rigorous way. The proposed approach (conformal alignment) can be applied to a wide range of real-life scenarios, e.g. medical decision-making (radiology report generating) and conversational chatbot (question answering).

## A    Deferred theory and proofs

### A.1    Proof of Theorem 3.1

Let $V(x, a) = 2\bar{M} \cdot \mathbb{1}\{a > c\} - g(x)$; define also $V_i = V(X_i, A_i)$ and $\widehat{V}_i = V(X_i, c)$ for every $i \in [n + m]$. By construction, $\widehat{V}_i = -\widehat{A}_i$; $V_i$ equals to $-\widehat{A}_i$ when $A_i \leq c$ and equals to $2\bar{M} - \widehat{A}_i$ otherwise. The conformal p-value defined in (3.2) can be written as

$$p_j = \frac{1 + \sum_{i \in \mathcal{D}_{\text{cal}}} \mathbb{1}\{A_i \leq c, \widehat{A}_i \geq \widehat{A}_{n+j}\}}{1 + |\mathcal{D}_{\text{cal}}|} = \frac{1 + \sum_{i \in \mathcal{D}_{\text{cal}}} \mathbb{1}\{V_i \leq \widehat{V}_{n+j}\}}{1 + |\mathcal{D}_{\text{cal}}|},$$

which is the p-value considered in [18]. It can also be checked that $\{V_i\}_{i \in \mathcal{D}_{\text{cal}}} \cup \{\widehat{V}_{n+j}\}_{j \in [m]}$ have not ties almost surely. Applying Theorem 2.6 of [18] concludes the proof.

### A.2    Proof of Proposition 3.3

The proof is an adaptation of Proposition 2.10 of [18]. We first note that the proof of Proposition 2.10 goes through with $U = 1$, which is the version we shall use in what follows.

To start, as in the proof of Theorem 3.1, we take $V(x, a) = 2M \cdot \mathbb{1}\{a > c\} - g(x)$. Let $F(v) = \mathbb{P}(V(X, A) \leq v)$ for any $v \in \mathbb{R}$. Then

$$F(v) = \mathbb{P}(V(X, A) \leq v) = \mathbb{P}(A \leq c, g(X) \geq -v) + \mathbb{P}(A < c, 2M - g(X) \leq v)$$
$$= H(-v) + \mathbb{P}(A < c, 2M - g(X) \leq v).$$

As a result, $F(V(X, c)) = F(-g(X)) = H(g(X))$, and for any $t \in \mathbb{R}$,

$$\frac{t}{\mathbb{P}(F(V(X, c)) \leq t)} = \frac{t}{\mathbb{P}(H(g(X)) \leq t)}.$$

Invoking Proposition 2.10 of [18], we have

$$\lim_{|\mathcal{D}_{\text{cal}}|, m \to \infty} \text{Power} = \frac{\mathbb{P}(F(V(X, c)) \leq t(\alpha), A > c)}{\mathbb{P}(A > c)}$$
$$= \frac{\mathbb{P}(H(g(X)) \leq t(\alpha), A > c)}{\mathbb{P}(A > c)}$$
$$= \mathbb{P}(H(g(X)) \leq t(\alpha) \,|\, A > c).$$

Similarly,

$$\lim_{|\mathcal{D}_{\text{cal}}|, m \to \infty} \frac{\sum_{j=1}^m \mathbb{1}\{j \in \mathcal{S}, A_{n+j} > c\}}{m} = \lim_{|\mathcal{D}_{\text{cal}}|, m \to \infty} \text{Power} \cdot \frac{\sum_{j=1}^m \mathbb{1}\{A_j > 0\}}{m}$$
$$= \mathbb{P}(H(g(X)) \leq t(\alpha) \,|\, A > c) \cdot \mathbb{P}(A > c)$$
$$= \mathbb{P}(H(g(X)) \leq t(\alpha), A > c).$$

We have therefore completed the proof.

### A.3    Finite-sample power analysis

In this part, we provide a finite-sample power analysis based on concentration inequalities. We define

$$F_+(t) = \mathbb{P}(g(X) \geq t, A > c), \quad F_-(t) = \mathbb{P}(g(X) \geq t, A \leq c), \quad F(t) = \mathbb{P}(g(X) \geq t).$$

The empirical version of these quantities are defined as

$$\widehat{F}_m^+(t) = \frac{1}{m}\sum_{j=1}^m \mathbb{1}\{g(X_{n+j}) \geq t, A_{n+j} > c\}, \quad \widehat{F}_n^+(t) = \frac{1}{n}\sum_{i=1}^n \mathbb{1}\{g(X_i) \geq t, A_i > c\},$$

$$\widehat{F}_m^-(t) = \frac{1}{m}\sum_{j=1}^m \mathbb{1}\{g(X_{n+j}) \geq t, A_{n+j} \leq c\}, \quad \widehat{F}_n^-(t) = \frac{1}{n}\sum_{i=1}^n \mathbb{1}\{g(X_i) \geq t, A_i \leq c\},$$

$$\widehat{F}_m(t) = \frac{1}{m}\sum_{j=1}^m \mathbb{1}\{g(X_{n+j}) \geq t\}, \quad \widehat{F}_n(t) = \frac{1}{n}\sum_{i=1}^n \mathbb{1}\{g(X_i) \geq t\},$$

We also define $\widehat{m}^+ = \sum_{j=1}^m \mathbb{1}\{A_{n+j} > c\}$ as the number of truly aligned test outputs.

**Proposition A.1.** Under the same condition of Theorem 3.1, assume further that $(X_i, E_i)_{i \in [n+m]}$ are i.i.d. For any $\delta \in (0,1)$, with probability at least $1 - \delta$,

$$\frac{F_+(\tau_-) - \epsilon_m}{\mathbb{P}(A > c) + \epsilon_m} \leq \frac{\sum_{j=1}^m \mathbb{1}\{g(X_{n+j}) \geq \widehat{\tau}, A_{n+j} > c\}}{\sum_{j=1}^m \mathbb{1}\{A_{n+j} > c\}} \leq \frac{F_+(\tau_+) + \epsilon_m}{\mathbb{P}(A > c) - \epsilon_m},$$

where

$$\tau_+ := \inf\left\{t \colon \frac{1}{n+1} \cdot \frac{1 + nF_+(t) + n\epsilon_n}{F_m(t) - \epsilon_m} \leq \alpha\right\}, \quad \tau_- := \inf\left\{t \colon \frac{1}{n+1} \cdot \frac{1 + nF_+(t) - n\epsilon_n}{F_m(t) + \epsilon_m} \leq \alpha\right\},$$

and $\epsilon_m = \sqrt{\log(8/\delta/(2n))}, \epsilon_n = \sqrt{\log(8/\delta)/(2m)}$.

*Proof of Proposition A.1.* Fix any constant $\delta \in (0,1)$. Consider the function $\widetilde{g}(X, A) = M \cdot \mathbb{1}\{A > c\} + g(X)$, where $M > 2\sup_{x \in \mathcal{X}} |g(x)| < \infty$ is a sufficiently large constant such that $\inf_{x \in \mathcal{X}} \widetilde{g}(x, c+1) > \sup_{x \in \mathcal{X}} \widetilde{g}(x, c)$. Therefore, we know that for any $|t| < M/2$, it holds that

$$F_+(t) = \mathbb{P}(\widetilde{g}(X, A) \geq M + t), \quad F_-(t) = \mathbb{P}(t \leq \widetilde{g}(X, A) < M/2).$$

And similar relations hold for the empirical versions of these quantities. Applying the Dvoretzky–Kiefer–Wolfowitz (DKW) inequality [10, 31] with a union bound for the empirical c.d.f. of $\widetilde{g}(X, A)$ for both calibration and test data, we know that with probability at least $1 - \delta$,

$$\sup_t \left|\widehat{F}_m^+(t) - F_+(t)\right| \leq \epsilon_m, \quad \sup_t \left|\widehat{F}_m^-(t) - F_-(t)\right| \leq \epsilon_m, \quad \sup_t \left|\widehat{F}_m(t) - F(t)\right| \leq \epsilon_m,$$

$$\sup_t \left|\widehat{F}_n^+(t) - F_+(t)\right| \leq \epsilon_n, \quad \sup_t \left|\widehat{F}_n^-(t) - F_-(t)\right| \leq \epsilon_n, \quad \text{and} \quad \left|\widehat{m}^+ - \mathbb{P}(A > c)\right| \leq \epsilon_m,$$

where recall that $\epsilon_m = \sqrt{\log(8/\delta)/(2m)}$ and $\epsilon_n = \sqrt{\log(8/\delta)/(2n)}$. We call the event that all above holds the "good event" $E^*$, which happens with probability at least $1 - \delta$.

By definition, we can show that the selection set of Conformal Alignment is equivalently $\mathcal{S} = \{j \in [m]\colon g(X_{n+j}) \geq \widehat{\tau}\}$, where $\widehat{\tau} = \inf\{t \colon \widehat{\text{FDR}}(t) \leq \alpha\}$, and

$$\widehat{\text{FDR}}(t) = \frac{m}{n+1} \cdot \frac{1 + \sum_{i=1}^n \mathbb{1}\{g(X_i) \geq t, A_i \leq c\}}{\sum_{j=1}^m \mathbb{1}\{g(X_{n+j}) \geq t\}} = \frac{1}{n+1} \cdot \frac{1 + n \cdot \widehat{F}_n^+(t)}{\widehat{F}_m(t)}.$$

By the DKW inequality above, with probability at least $1 - \delta$,

$$\frac{1}{n+1} \cdot \frac{1 + nF_+(t) - n\epsilon_n}{F_m(t) + \epsilon_m} \leq \widehat{\text{FDR}}(t) \leq \frac{1}{n+1} \cdot \frac{1 + nF_+(t) + n\epsilon_n}{F_m(t) - \epsilon_m}$$

holds simultaneously for all $t \in \mathbb{R}$. On the other hand, the (realized) power is

$$\text{Pow} := \frac{\sum_{j=1}^m \mathbb{1}\{g(X_{n+j}) \geq \widehat{\tau}, A_{n+j} > c\}}{\sum_{j=1}^m \mathbb{1}\{A_{n+j} > c\}} = \frac{\widehat{F}_m^+(\widehat{\tau})}{m^+}.$$

On the good event $E^*$, we have

$$\frac{F_+(\widehat{\tau}) - \epsilon_m}{\mathbb{P}(A > c) + \epsilon_m} \leq \text{Pow} \leq \frac{F_+(\widehat{\tau}) + \epsilon_m}{\mathbb{P}(A > c) - \epsilon_m}.$$

By the definition of $\widehat{\tau}$, we know that $\tau_- \leq \widehat{\tau} \leq \tau_+$ on the good event $E^*$. The monotonicity of realized power as a function of selection cutoff $\widehat{\tau}$ thus implies that

$$\text{Pow} \geq \frac{\widehat{F}_m^+(\tau_-)}{m^+} \geq \frac{F_+(\tau_-) - \epsilon_m}{\mathbb{P}(A > c) + \epsilon_m},$$

$$\text{Pow} \leq \frac{\widehat{F}_m^+(\tau_+)}{m^+} \leq \frac{F_+(\tau_+) + \epsilon_m}{\mathbb{P}(A > c) - \epsilon_m}.$$

$\square$

# B   Additional comparison with prior works

## B.1   Comparison with selective prediction

For selective prediction, El-Yaniv et al. [11, 12] suggest considering the *risk* measure, defined (in our context) as:

$$R := \frac{\mathbb{E}\big[\sum_{j \in [m]} \mathbb{1}\{A_{n+j} \leq c, j \in \mathcal{S}\}\big]}{\mathbb{E}\big[\sum_{j \in [m]} \mathbb{1}\{j \in \mathcal{S}\}\big]}.$$

This measure is known as the marginal FDR (mFDR) in the multiple testing literature and there has been an exhaustive discussion of its comparison with FDR. For example, Benjamini and Hochberg [5] proposed FDR in favor of mFDR, since the latter is impossible to control in the global null case; Storey [45] also pointed out that mFDR cannot be used for controlling the joint behavior of the numerator (number of false selections) and the denominator (number of selections). On the other hand, to control the risk/mFDR, [12] proposes a method that searches for the cutoff over a grid of values, and then takes a union bound to achieve the overall error control, which might be conservative. In contrast, our method for FDR control does not involve taking union bounds, delivering sharp finite-sample control guarantees.

## B.2   Comparison with conformalized selection

Our work instantiates conformalized selection [18] for the reliable deployment of foundation model outputs. We detail our contributions as follows. First, we justify that selecting reliable outputs with FDR control is a practically reasonable criterion for downstream tasks. Second, we formulate the alignment problem as a selective prediction problem that can be tackled with conformal selection, adapting conformal selection to the current setting. Note that while conformal selection is an established method, its current application in job hiring and drug discovery requires numerical outputs and predictions; the task of aligning foundation model outputs, however, does not come with a clear numerical output. To make conformal selection applicable to our purpose, we formulate the alignment status as a numerical indicator that could be inferred with reference information, and then decide how to train a prediction model for the alignment score, which can therefore be used in conjunction with the conformal selection framework. Our third contribution is the implementation of the pipeline. In this regard, we conduct extensive numerical experiments, finding that using a lightweight prediction model such as logistic regression or random forests with popular heuristic (uncalibrated) uncertainty measures is often sufficient for identifying reliable outputs. This provides a concrete and easy-to-use pipeline for practical use.

# C   Additional implementation details

This section contains implementation details for Section 4 and 5.

## C.1 Question answering

As is mentioned in Section 4, we adopt a subset of sample size 8000 from the TriviaQA dataset [21][5] and a subset of sample size 7700 from the CoQA dataset [39][6]. For both TriviaQA and CoQA datasets, we follow the exact text preparation and prompt engineering procedure in [29], which we show below. To evaluate the correctness of generated outputs, we adopt the `rouge-L` score [28] that focuses on the longest common subsequence between two sequences. Following the practice in [24, 29], a generated answer is defined to be admissible when the `rouge-L` score is no less than $0.3$.

**TriviaQA.** The TriviaQA dataset can be loaded from the Python module `datasets` and we use the `rc.nocontext` split. Prompts in use are the same as those in [29, 47] and take the following format.

```
Answer these questions:
Q: In Scotland a bothy/bothie is a?
A: House
Q: [sample question]
A:
```

**CoQA.** The CoQA dataset is downloaded from `https://downloads.cs.stanford.edu/nlp/data/coqa/coqa-dev-v1.0.json` and the prompts are the same as those in [29, 24].

```
[The provided context paragraph]
[additional question-answer pairs]
Q: [question].  A:
```

Answers are generated using the default configurations similar to [29]; in specific, we use `num_beams=1, do_sample=True, top_p=1.0, top_k=0, temperature=1.0`. For each question, we generate $M = 20$ answers independently, with which we treat the first output as $f(X_i)$ and utilize the $M$ outputs to calculate the similarity-base scores.

Given the generated outputs, we adopt the pipeline in [29] to calculate the uncertainty and confidence scores. The self-evaluation score (`Self_Eval`), or `P(True)`, is obtained using the same prompt in [22] as follows:

```
[story]
Question:  [question]
Here are some brainstormed ideas:  [few_shots examples]
Possible Answer:  [generated answer] Is the possible answer:
(A) True
(B) False
The possible answer is:  (
```

The self-evaluation score is then defined as the output probability associated with the generated answer "A)".

**Language models.** We use two language models without fine-tuning for comparison in terms of the accuracy of the generated answers: OPT-13B and LLaMA-2-13B-chat. The implemented OPT-13B model is from Hugging Face `https://huggingface.co/facebook/opt-13b` and the implemented LLaMA-2-13B-chat is from `https://llama.meta.com`. We utilize an off-the-shelf DeBERTa-large model [13][7] as the NLI classifier to calculate similarities.

---

[5] `https://nlp.cs.washington.edu/triviaqa/`
[6] `https://stanfordnlp.github.io/coqa/`
[7] `https://github.com/microsoft/DeBERTa`

## C.2 CXR report generation

We use the subfolders `p10,p11,p12` from the MIMIC-CXR dataset [20] (these are all the data we use, including fine-tuning and subsequent experiments for Conformal Alignment) to limit the resources required for fine-tuning. The MIMIC-CXR dataset can be accessed from the PhysioNet project page https://physionet.org/content/mimic-cxr/2.0.0/ under the PhysioNet Credentialed Health Data License 1.5.0.

In this application, we fine-tune a vision-language model, with the Vision Transformer `google/vit-base-patch16-224-in21k` pre-trained on ImageNet-21k[8] as the image encoder and GPT2 as the text decoder. We largely follow the training procedure in [36]. In particular, each raw image is resized to $224 \times 224$ pixels. We then fine-tune the model on a hold-out dataset with a sample size of $43,300$ for 10 epochs with a batch size of 8, and other hyperparameters are set to default values. The training process takes about 10 hours on one NVIDIA A100 GPU.

Once fine-tuned, the model is treated as fixed for generating reports. The data not used in fine-tuning are used as the training, calibration, and test data in our experiments. For report generation, we use the default configurations (`max_length=512, do_sample=True`) and the input of the vision-language model consists of pixel values of images (resized to $224 \times 224$ pixels) and tokenized prompts (tokenized via the same GPT2 model). For each image, we generate $M = 10$ reports independently, with which we treat the first output as $f(X_i)$ and utilize the $M$ outputs to calculate the features (similarity-based scores) for training the alignment predictor. To determine if a generated report is admissible, we follow the practice in [36] and use the CheXbert model [44] to convert both the generated and reference reports into 14-dimensional vectors. For each coordinate, the entry falls into the categories "Blank", "Positive", "Negative" or "Uncertain", with which we further convert them to binary vectors with each entry indicating "Positive" or not. The alignment score $A_i \in \{0, 1\}$ is defined to be the indicator of whether there are at least 12 matched coordinates between the two binary vectors. Similar to the QA task, we modify the pipeline in [29] to calculate the uncertainty and confidence scores, in which we utilize an off-the-shelf DeBERTa-large model [13] as the NLI classifier to calculate similarities.

# D  Additional experimental results for question answering

## D.1  FDR/Power for TriviaQA dataset with alternative classifiers

We present the FDR/power plots when applying Conformal Alignment to the TriviaQA dataset and the alignment predictor is trained with random forests (Figure 8) and XGBoost (Figure 9), fixing $\gamma_1 = 0.2$ and $\gamma_2 = 0.5$. These figures are similar to Figure 3 (which uses logistic regression to train the alignment predictor) in the main text. We observe similar messages as those in Section 4: the FDR is always tightly controlled with satisfactory power. Regarding the choice of $|\mathcal{D}|$, we similarly see that 500 high-quality samples suffice for the powerful deployment of our method.

---

[8]https://huggingface.co/google/vit-base-patch16-224-in21k

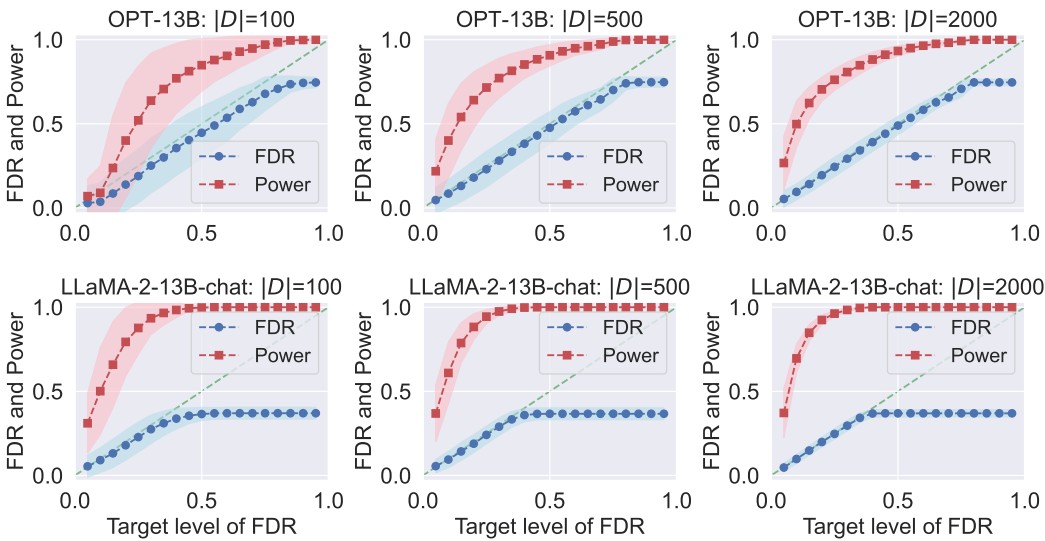

Figure 8: FDR/power plots for conformal alignment with TriviaQA dataset and random forests as the base classifier. Details are otherwise the same as Figure 3.

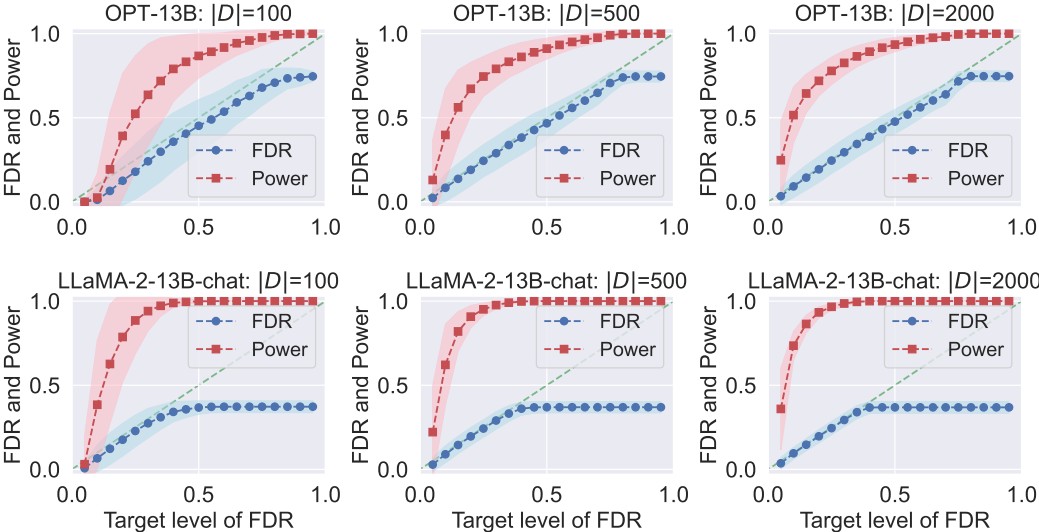

Figure 9: FDR/power plots for conformal alignment with TriviaQA dataset and XGBRF as the base classifier. Details are otherwise the same as Figure 3.

## D.2   FDR/Power on CoQA dataset

In this part, we present a set of results parallel to the main text and Section D.1 when we apply Conformal Alignment to the CoQA dataset; details about the dataset are in Appendix C.1.

We present the FDR/power plots for the CoQA dataset when the alignment predictor is trained with logistic regression (Figure 10), random forest (Figure 11), and XGBoost (Figure 12), fixing $\gamma_1 = 0.2$ and $\gamma_2 = 0.5$. We again observe similar messages as those in Section 4: the FDR remains tightly controlled, and a sample size of $|\mathcal{D}| = 500$ suffices for the powerful deployment of our method.

We also recall the baseline, where one asks the language model to evaluate its confidence (i.e., the `Self_Eval` feature above), and threshold the obtained likelihood $\{q_j^{\text{self}}\}$ with a cutoff of $1 - \alpha$ as if it were the "probability" of alignment, i.e., $\mathcal{S}_{\text{baseline}} = \{j \in [m] : q_j^{\text{self}} \geq 1 - \alpha\}$. In Figure 13, we

present the results with the CoQA dataset, in which the baseline fails to control FDR for any given $\alpha$ and also exhibits lower power than our proposed approach.

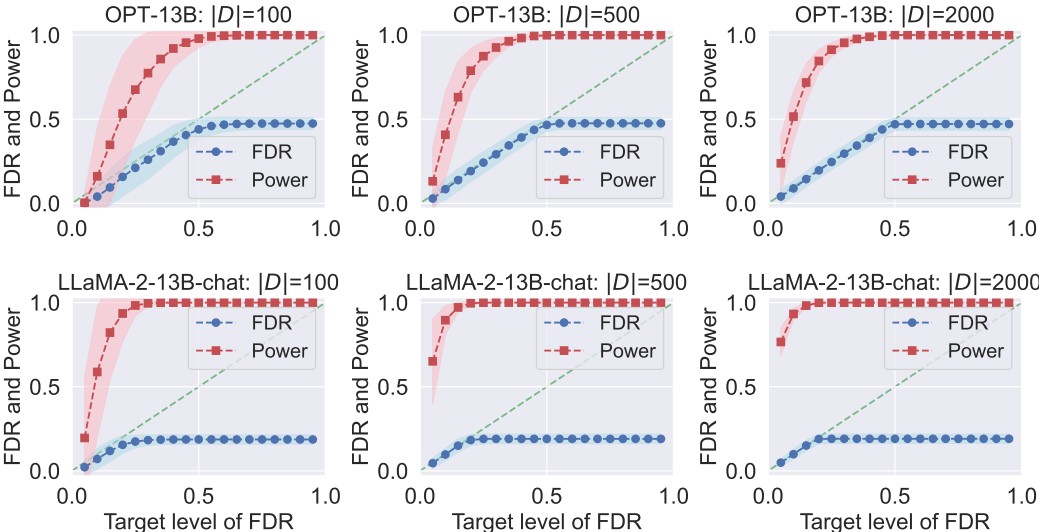

Figure 10: FDR/power plots for conformal alignment with CoQA dataset and logistic regression for training the alignment predictor. Details are otherwise the same as Figure 3.

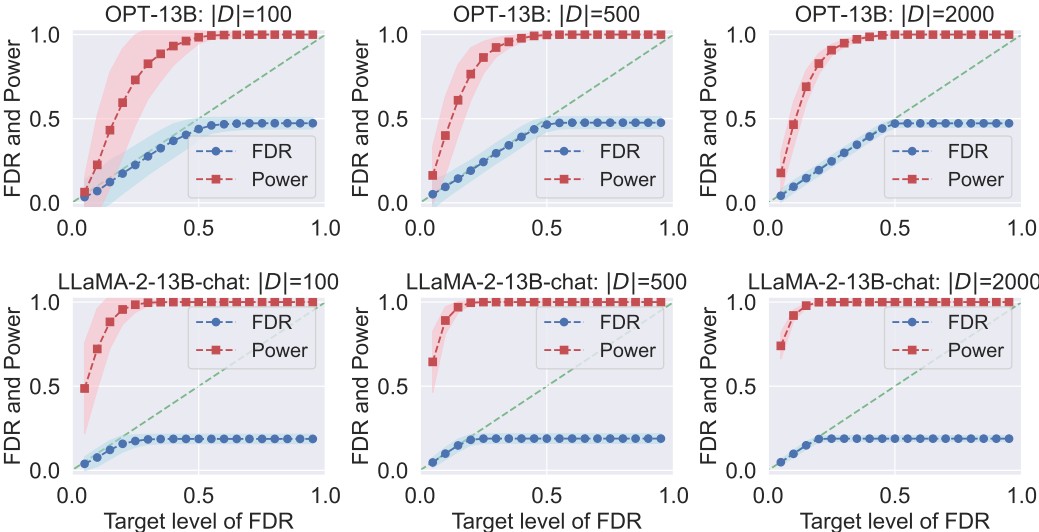

Figure 11: FDR/power plots for conformal alignment with CoQA dataset and random forests as the base classifier. Details are otherwise the same as Figure 3.

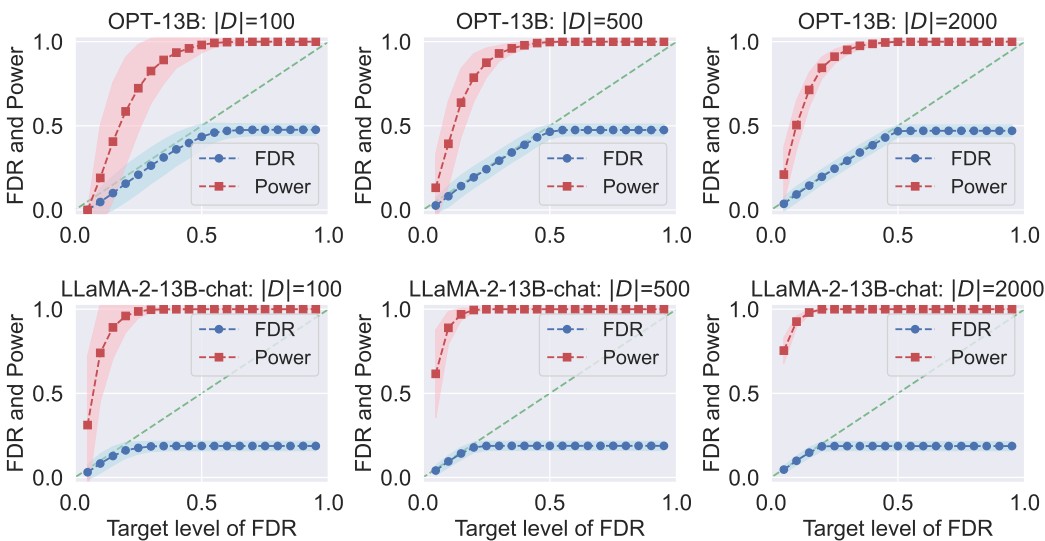

Figure 12: FDR/power plots for conformal alignment with CoQA dataset and XGBRF as the base classifier. Details are otherwise the same as Figure 3.

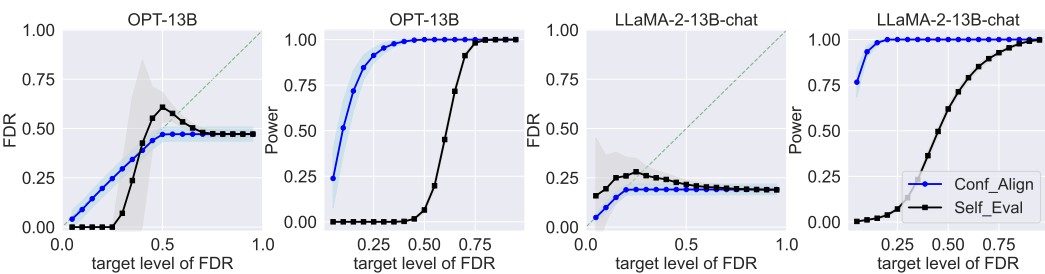

Figure 13: Comparison of Conformal Alignment at FDR level $\alpha$ versus the heuristic baseline applied to CoQA dataset. Details are otherwise the same as Figure 5.

### D.3 FDR for individual features

Figure 14 presents the realized FDR at each FDR target level for the TriviaQA dataset when the alignment predictor is a logistic regression over each individual feature.

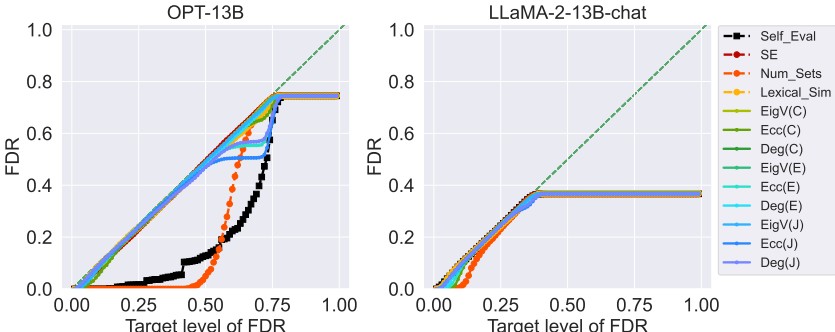

Figure 14: The realized FDR at each FDR target level for TriviaQA dataset when the alignment predictor is trained with logistic regression over each individual feature.

From Figure 14, we note that FDR is always controlled for each feature. However, FDR with more powerful LLM (LLaMA-2-13B-chat) is tightly controlled while `Self_Eval` and `Num_Sets` with OPT-13B produce FDR and power both close to zero when $\alpha$ is small as their predictive power is too low. We omit similar results when conducting the same experiment on the CoQA dataset.

### D.4  Ablation studies

In this part, we present and discuss ablation studies for choosing the data splitting schemes $\gamma_1, \gamma_2$ through experiments on the TriviaQA dataset, with the alignment predictor trained via logistic regression over all the features in the main text.

**Varying $\gamma_1$ (size of the tuning set).**    Recall that we use a random subset of size $\lfloor \gamma_1 |\mathcal{D}| \rfloor$ for tuning hyperparameters [29] in the features, and a subset of size $\lfloor \gamma_2 |\mathcal{D}| \rfloor$ for training the alignment predictor. We start by studying the effect of $\gamma_1$ on the performance with fixed $\gamma_2$. This means the sample size for predictor training is fixed, but there is a tradeoff between the sample sizes used for hyperparameter tuning and the calibration step. Here, logistic regression is used to train the alignment predictor and the total number of "high-quality" data is fixed at $|\mathcal{D}| = 2000$.

We show the results when we vary $\gamma_1$ in $\{0.2, 0.4, 0.6\}$ for the TriviaQA dataset (Figure 15) and the CoQA dataset (Figure 16).

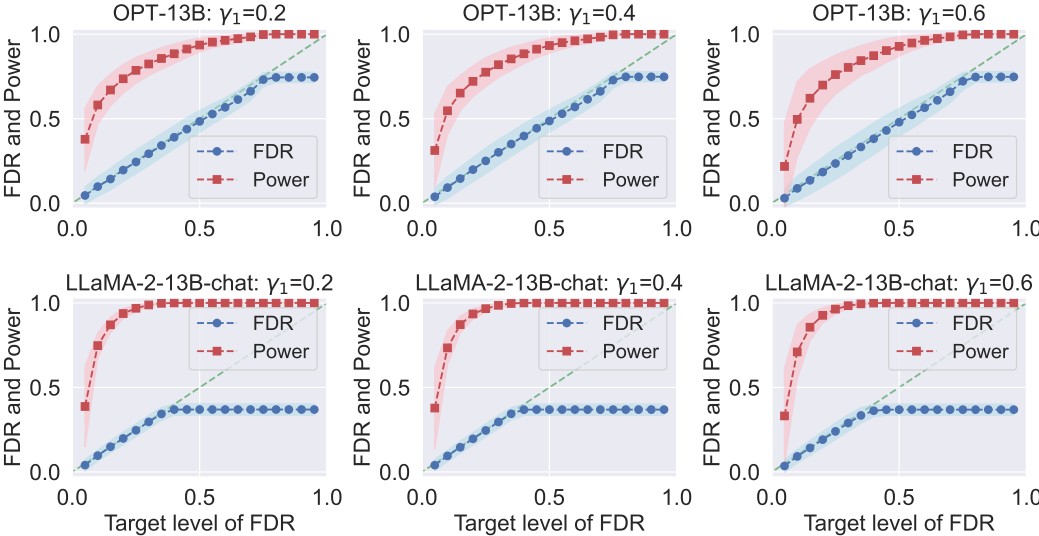

Figure 15: Ablation study for the choice of $\gamma_1$: FDR/power plots for conformal alignment with TriviaQA dataset. We fix $|\mathcal{D}| = 2000$, $\gamma_2 = 0.3$, and use logistic regression as the base classifier.

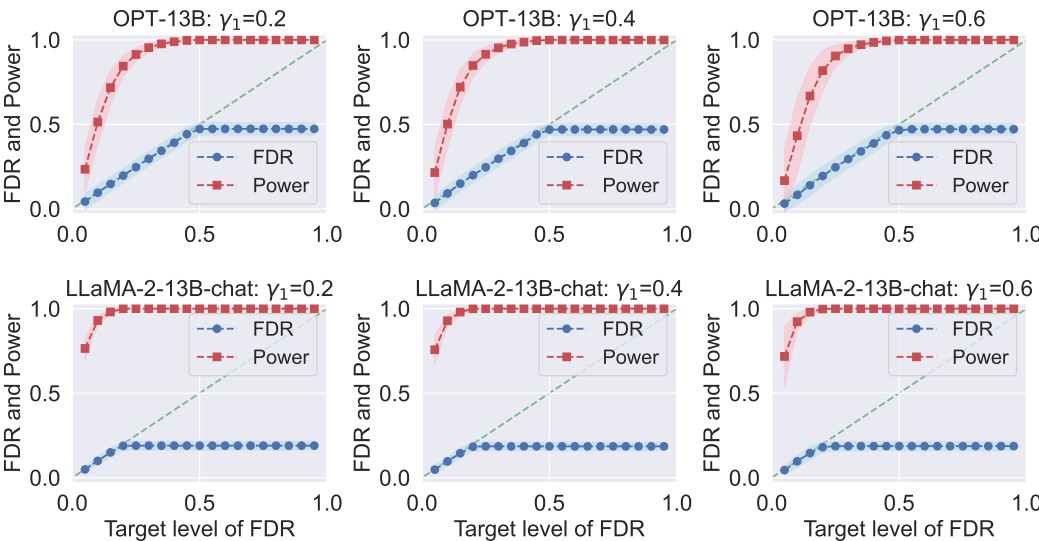

Figure 16: Ablation study for the choice of $\gamma_1$: FDR/power plots for conformal alignment with TriviaQA dataset: $|\mathcal{D}| = 2000$, $\gamma_2 = 0.3$, logistic regression as the base classifier.

In Figure 16, we can see that with fixed $|\mathcal{D}|$ and $\gamma_2$, when $\gamma_1$ increases, the power slightly decreases due to the decreasing sample size for training the alignment score predictor and BH procedure. Besides, we note that $\gamma_1 = 0.2$ is sufficient for tuning the parameters to calculate the features.

**Varying $\gamma_2$ (size of the predictor training set).** We proceed to study the choice of $\gamma_2$, the proportion of reference data used for training the classifier. We vary the value of $\gamma_2$ while fixing the size of the reference set $|\mathcal{D}| = 2000$ and $\gamma_1 = 0.2$ per our recommendation from the last part. Note that with $|\mathcal{D}|$ and $\gamma_1$ fixed, varying $\gamma_2$ also reveals the effect of $|\mathcal{D}_{\mathrm{cal}}|$. We use logistic regression as the base classifier. The results are shown for TriviaQA dataset in Figure 17 and CoQA dataset in Figure 18.

At values $\gamma_2 = 0.1$ and $\gamma_2 = 0.7$, we observe a slight power loss compared with the middle two columns. The reason is that when $\gamma_2$ is close to either $0$ or $1$, there will be unbalanced sample sizes between $\mathcal{D}_{\mathrm{tr}}^{(2)}$ and $\mathcal{D}_{\mathrm{cal}}$, and the scarce of sample in each set will affect the performance, i.e. causing larger variance in FDR/power or smaller power. In light of this, we suggest the choice of $\gamma_2 \in [0.3, 0.5]$ to balance $\mathcal{D}_{\mathrm{tr}}^{(2)}$ and $\mathcal{D}_{\mathrm{cal}}$.

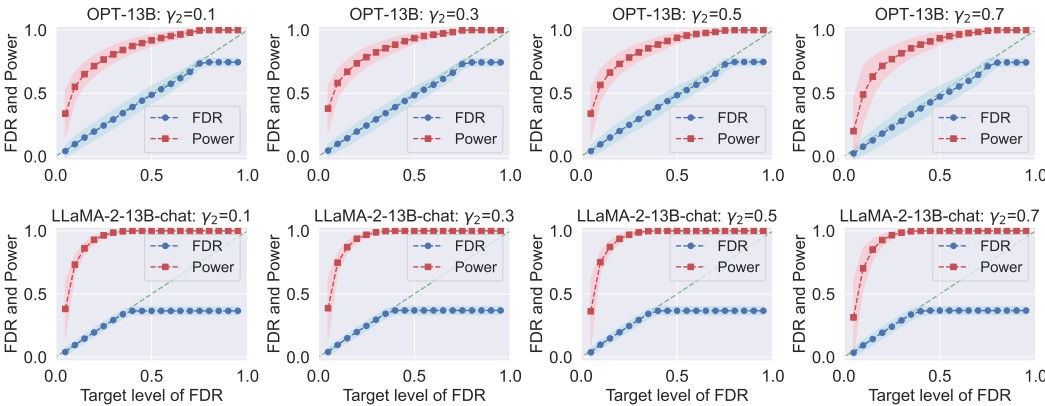

Figure 17: Ablation study for the choice of $\gamma_2$: FDR/power plots for conformal alignment with TriviaQA dataset, fixing $|\mathcal{D}| = 2000$, $\gamma_1 = 0.2$, and using logistic regression as the base classifier.

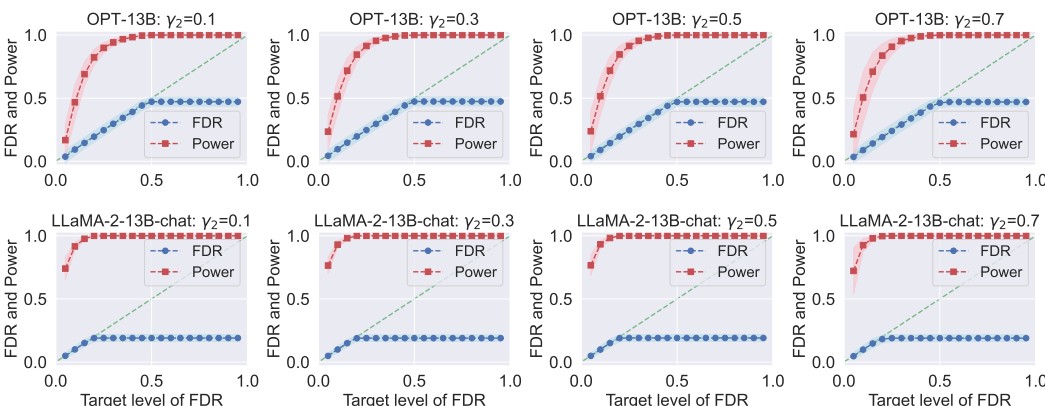

Figure 18: Ablation study for the choice of $\gamma_2$: FDR/power plots for conformal alignment with CoQA dataset, fixing $|\mathcal{D}| = 2000$, $\gamma_1 = 0.2$, and using logistic regression as the base classifier.

| Question | Reference answer | Generated answer | Alignment scores |
|---|---|---|---|
| Which company produced the Hastings and Herald aircraft? | Handley-Page | de Havilland | $A_i = 0$ 
 $\widehat{A}_i = 0.083$ |
| How many 'E' tiles are provided in a Scrabble game? | 12 | 2 | $A_i = 0$ 
 $\widehat{A}_i = 0.190$ |
| In the 1972 film Cabaret, Sally Bowles is working in which club? | KitKat | Kit Kat Klub | $A_i = 0$ 
 $\widehat{A}_i = 0.505$ |
| An orrery, popular in the 13th and 19th centuries, was a model of what? | The Solar System | Solar system | $A_i = 1$ 
 $\widehat{A}_i = 0.814$ |
| Of which band was Feargal Sharkey the lead singer until 1983? | THE UNDERTONES | The Undertones | $A_i = 1$ 
 $\widehat{A}_i = 0.938$ |

Table 1: Illustration on TriviaQA dataset with LLaMA-2-13B-chat: selection threshold $\widehat{\tau} = 0.388$.

## D.5 Real examples in QA

## D.6 Empirical evaluation of feature importance

In Figure 5 and 7, we quantify the importance of individual features in the alignment predictor via the ROC curve. In this section, we use the Shapley value as well as the model-based score to evaluate the individual feature importance more directly.

We used the XGBoost functionality to compute the importance scores in Figure 19d. We also computed the Shapley value of individual features to measure their importance based on the prediction models we use in the manuscript. As the Shapley values are model-agnostic, we present results for three choices of $g(X)$ with different base classifiers, including logistic regression, random forests, and XGBoostRF, in Figure 19a, 19b, and 19c. Although the exact order of features varies in four plots, there are features that have consistent dominating effects in all figures, e.g. Deg(J) and EigV(J) based on the Jaccard similarity. In addition, NumSets as the feature in alignment score predictors tends to exhibit an effect close to zero in all four figures. We should note that the aforementioned

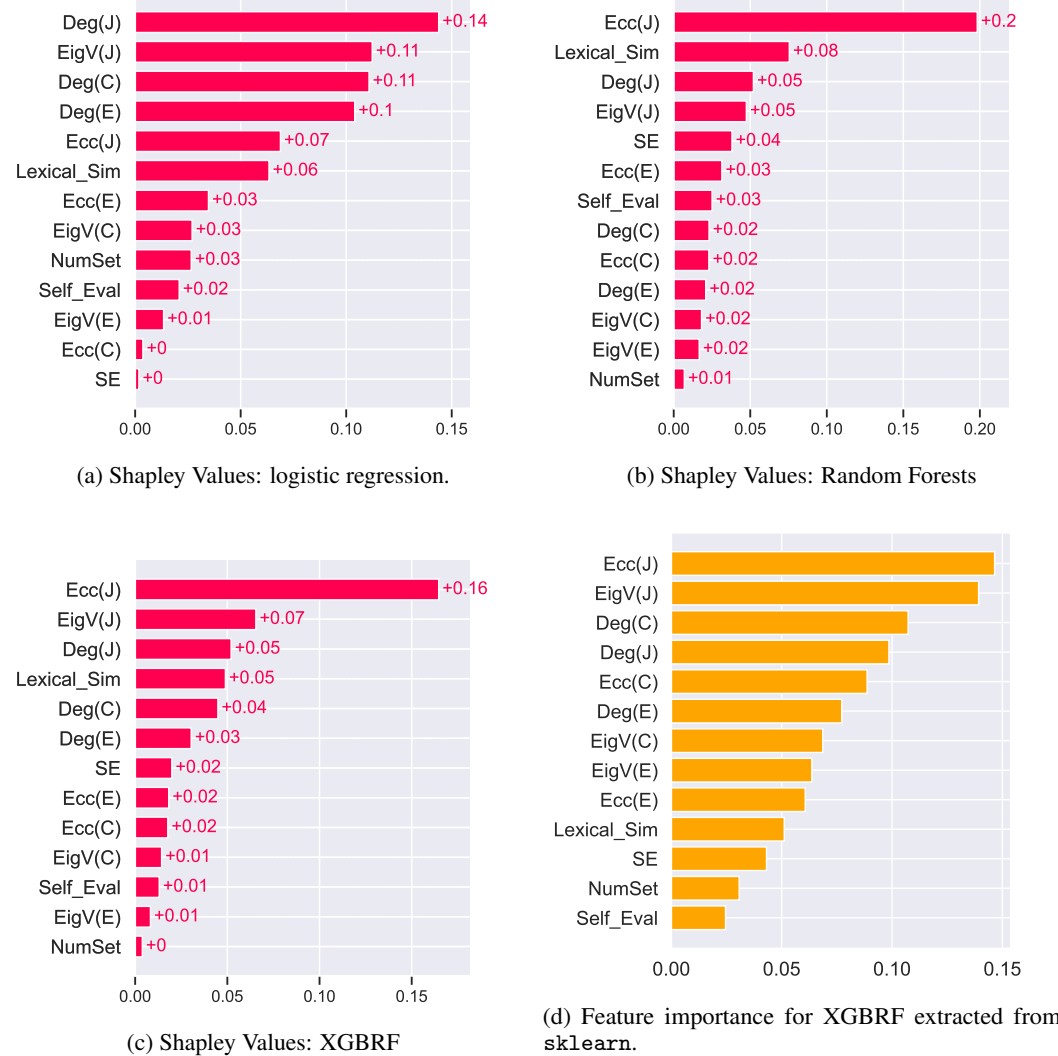

(a) Shapley Values: logistic regression.

(b) Shapley Values: Random Forests

(c) Shapley Values: XGBRF

(d) Feature importance for XGBRF extracted from `sklearn`.

Figure 19: Measures of Feature importance (Dataset: TriviaQA, LLM: LLaMA-2-13B-chat).

findings are also revealed in Figure 5 in the paper, where predicted scores with `EigV(J), Deg(J)`, and `Ecc(J)` have higher power and that with `NumSets` is much less powerful. We thank the reviewer again for raising this question for more comparison of feature importance in the alignment score predictor.

### D.7 Comments on FDR guarantee and comparison to other frameworks

The major distinction between our method and other applications of conformal prediction on the foundation models is the type of guarantee we pursue. This is related to the following question: how can we use the outputs of these algorithms in practice? In short, (a) For [36], it is unclear how to pick one answer from the multiple answers in a way that does not hinder validity. (b) Conformal actuality [32] makes the original outputs less specific, which can hinder downstream use.

To demonstrate (a), we conducted additional experiments for more clear comparison. With the `TriviaQA` dataset and the base model `LLaMA-2-13B-chat`, we follow the procedure in [36] and have the following examples:

- Example 1: Question: In tennis, losing two sets 6-0 is known as a double what? True answer: Bagel. Generated set: {'bagel'}.

- Example 2: Question: What was the name of the brothel in The Best Little Whorehouse in Texas? True answer: Chicken ranch. Generated set: {'miss monas', 'chicken ranch', 'miss mona's', 'miss monas bordello'}.

As we can see in Example 1, the generated set is very informative and can be used directly. However, although the generated set in the second example has a high probability of covering the true answer, it contains very different answers and is potentially confusing in practice. In other words, whether "Chicken ranch" or "Miss Monas" is correct is unclear to the user.

In addition, as is shown in Figure 20 (left panel), we present the proportion that the size of generated size is $k$ for different values of $k \in \mathbb{N}$, which shows that there are more than $40\%$ of sets having no less than $4$ distinct items. In addition, Figure 20 (right panel) shows that in nearly $30\%$ of the questions, the admissible answers (with $\texttt{rougeL} \geq 0.3$) take up less than $20\%$ of the uncertain set, which poses a challenge in the direct deployment of answers in the uncertain set.

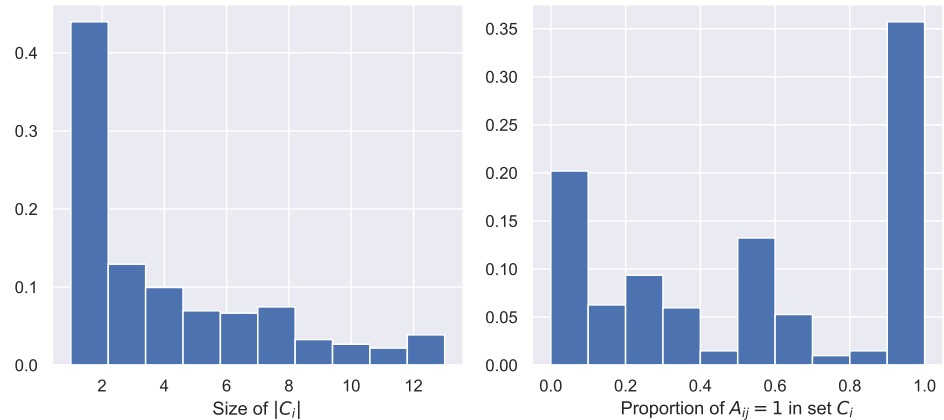

Figure 20: Additional histograms for Conformal Language Modeling: sizes of uncertainty sets and proportions of admissible answers, where $C_i$ denotes the uncertainty set for question No.$i$ (Dataset: TriviaQA, LLM: LLaMA-2-13B-chat).

# E    Additional experimental results for CXR report generation

## E.1    Additional results with alternative classifiers

We also use random forests and XGBRF as alternative classifiers to train the alignment score predictor. Similar to Section 5, we fix $\gamma_1 = 0.2$ and $\gamma_2 = 0.5$. The results are in Figure 21 for random forests, and Figure 22 for XGBoost. As $|\mathcal{D}|$ increases, we observe a slight increase as well as a decrease in variance in the power curve while FDR remains tightly controlled.

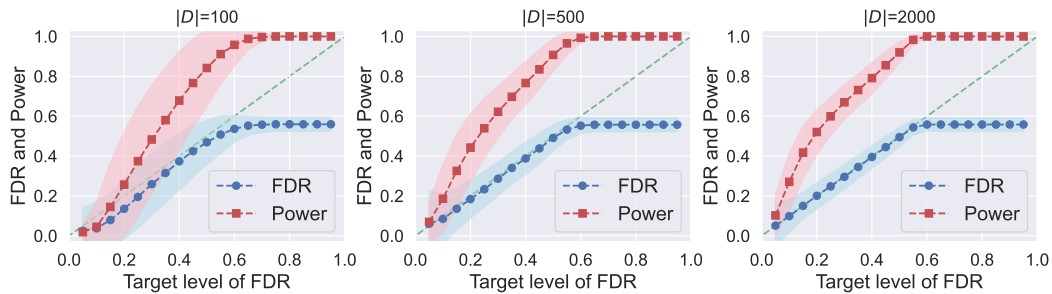

Figure 21: FDR/power plots for conformal alignment with CXR dataset when using random forests to train the alignment predictor. Details are otherwise the same as in Figure 6.

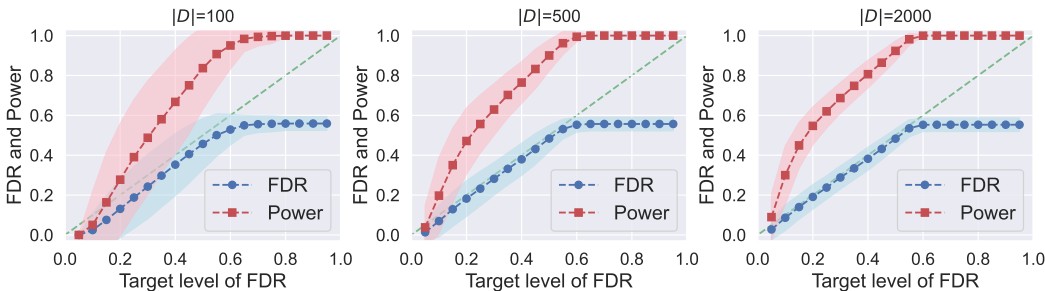

Figure 22: FDR/power plots for conformal alignment with CXR dataset when using XGBRF to train the alignment predictor. Details are otherwise the same as in Figure 6.

## E.2 Informativeness of individual features

Figure 23 shows the plots of FDR in the same experiment with Figure 7. Recall that we use one feature at a time to train the alignment score predictor. While FDR is controlled for all features, Num_Sets and Ecc(C) tend to return empty selection sets $\mathcal{S}$ when $\alpha$ is small, which results in zero FDR as well as zero power. Besides, Ecc(E) is also relatively less powerful, which implies that the context of CXR report can be challenging for the Eccentricity score since the dimension embeddings is much higher.

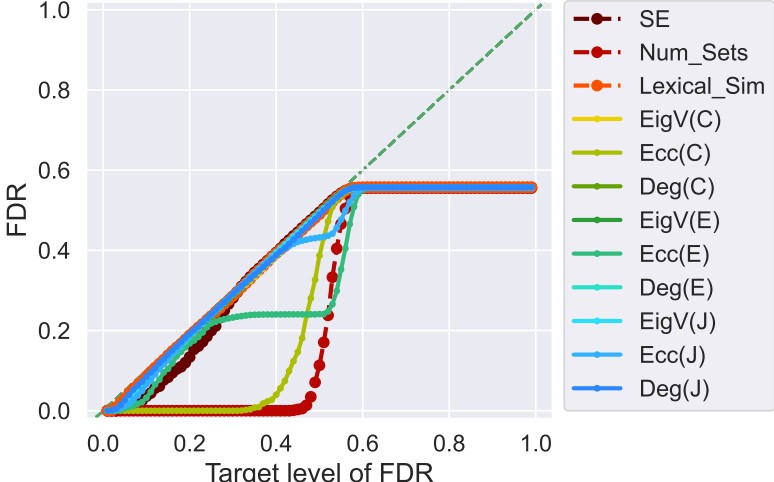

Figure 23: FDR at each FDR level for CXR dataset when the alignment predictor is trained with logistic regression over each individual feature, fixing $|\mathcal{D}| = 2000$, $\gamma_1 = 0.2$, $\gamma_2 = 0.5$.

## E.3 Ablation studies

We present ablation studies for the sample splitting hyperparameters $\gamma_1, \gamma_2$.

**Varying $\gamma_1$ (size of the tuning set).** With $|\mathcal{D}| = 2000$, $\gamma_2 = 0.3$ and logistic regression as the base classifier, we vary $\gamma_1$ in $\{0.2, 0.4, 0.6\}$ in Figure 24, in which both FDR and power exhibit larger variance when $\gamma_1$ is close to 1 due to limited sample size in $\mathcal{D}_{\text{tr}} \cup \mathcal{D}_{\text{cal}}$. Thus, we recommend $\gamma_1 = 0.2$ similar to the QA tasks.

**Varying $\gamma_2$ (size of the predictor training set).** With $|\mathcal{D}| = 2000$, $\gamma_1 = 0.2$, and logistic regression as the base classifier, the proportion of the training set varies in $\{0.1, 0.3, 0.5, 0.7\}$.

In Figure 25, the plots with $\gamma_2 = 0.1, 0.3, 0.5$ are comparable, but as $\gamma_2$ is close to 1, implying that the $\mathcal{D}_{\text{cal}}$ will produce p-values with a low resolution, the FDR and power are less stable. In this case, it reveals that there are different requirements in sample size for $\mathcal{D}_{\text{tr}}$ and $\mathcal{D}_{\text{cal}}$. To better understand

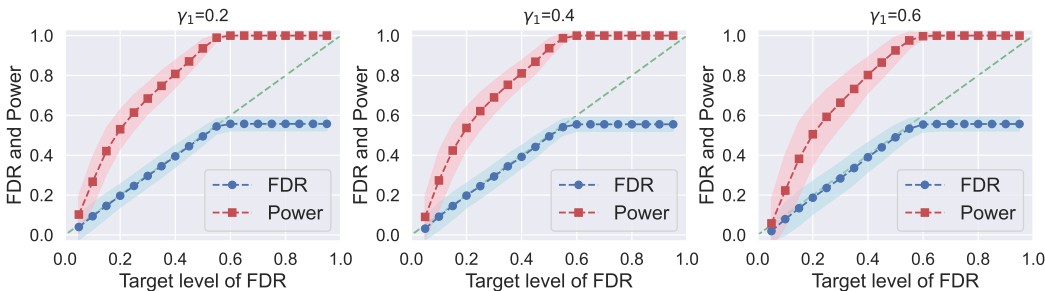

Figure 24: Ablation study for $\gamma_1$: FDR/power plots for conformal alignment with CXR dataset when fixing $|\mathcal{D}| = 2000$, $\gamma_2 = 0.3$ and using logistic regression to train the alignment predictor.

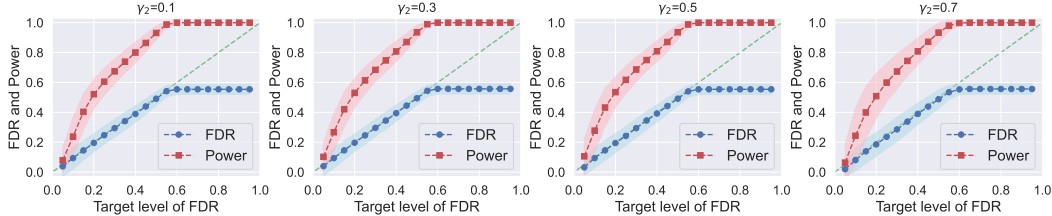

Figure 25: FDR/power plots for conformal alignment with CXR dataset: $|\mathcal{D}| = 2000$, $\gamma_1 = 0.2$, logistic regression as the base classifier.

the asymmetric roles of the training and calibration sets, we further decrease the size of the reference dataset and choose $|\mathcal{D}| = 100$ to repeat the experiment.

Figure 26 further shows that too small (0.1) or too large (0.7) proportion of $\mathcal{D}_{\mathrm{tr}}$ are both not preferred, thus any value $\gamma_2 \in [0.3, 0.5]$ is suggested.

### E.4 Real examples in report generation

To illustrate the role of the alignment score predictor, we present examples of CXR images with reference and generated reports in Table 2. In this realization, following Algorithm 1, the selected set is obtained by $\mathcal{S} = \{i \in \mathcal{D}_{\mathrm{cal}} : \widehat{A}_i \geq \widehat{\tau}\}$ with $\widehat{\tau} = 0.592$. For the first image, as highlighted in pink, the foundation model misreads the size of the cardiac silhouette. Accordingly, the alignment score predictor returns $\widehat{A}_i = 0.194 < \widehat{\tau}$, and the conformal alignment procedure successfully excludes this generated report. For the second image, the conditions of the cardio-mediastinal silhouette and hilar contours are correctly captured by the foundation model, and a correct assessment of both pleural effusion and pneumothorax is provided.

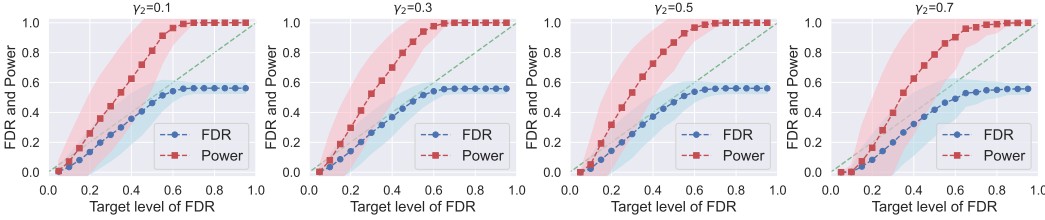

Figure 26: FDR/power plots for conformal alignment with CXR dataset: $|\mathcal{D}| = 100$, $\gamma_1 = 0.2$, logistic regression as the base classifier.

| CXR image | Reference report | Generated report |
|---|---|---|
| 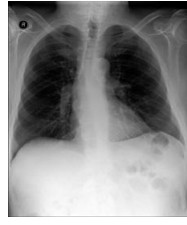

Alignment scores:
$A_i = 0$
$\widehat{A}_i = 0.194 < \widehat{\tau}$ | In comparison with the study of _, there has been a substantial increase in opacifications diffusely involving both lungs. `Cardiac silhouette remains within normal limits` and there is no evidence of pleural effusion.  The appearance suggests diffuse pulmonary edema.  However, in the appropriate clinical setting, widespread pneumonia or even ARDS could be considered. | In comparison with the study of _, there is little overall change.  Again `there is a substantial enlargement of the cardiac silhouette` with diffuse bilateral pulmonary opacifications consistent with pulmonary edema.  In the appropriate clinical setting, superimposed pneumonia would have to be considered. |
| Alignment scores:
$A_i = 1$
$\widehat{A}_i = 0.875 > \widehat{\tau}$ | The lungs are well inflated and clear.  `The cardiomediastinal silhouette, hilar contours`, and pleural surfaces `are normal.` There is `no pleural effusion or pneumothorax.` Mild degenerative changes are seen throughout the thoracic spine.  IMPRESSION: No acute cardiopulmonary process. | `The cardiomediastinal and hilar contours are within normal limits.` The lungs are clear `without` focal consolidation, `pleural effusion or pneumothorax`.  IMPRESSION: No acute cardiopulmonary process. |

Table 2: Illustration of conformal alignment with CXR examples: selection threshold $\widehat{\tau} = 0.592$.

