# OpenReview forum: "Conformal Alignment: Knowing When to Trust Foundation Models with Guarantees"
_NeurIPS.cc/2024/Conference — NeurIPS 2024 poster_

### Official Review · Reviewer_SzCp · 2024-07-09

**Soundness:** 4
**Presentation:** 3
**Contribution:** 3
**Rating:** 6
**Confidence:** 3

**Summary:**

The paper proposes to adapt conformal guarantees to large language models. They build upon existing work to provide bounds on FDR error for each model output by conformalizing the distribution of an additional alignment predictor trained on labeled aligned data. Experiments are conducted on two radiological report generation and question answering using pretrained LLM models.

**Strengths:**

* The paper addresses an important topic, is well-written, and effectively contextualizes related work
* The proposed method is easy drop-in for most systems and has wide utility in downstream applications
* The guarantee offered is applicable to free text outputs and flexible in types of alignment functions

**Weaknesses:**

* The technical novelty of the method seems limited as it seems to be a direct application of prior work for LLM's https://arxiv.org/abs/2210.01408 (more discussion of the differences would be helpful in clarifying technical novelty)
* There is a lack of comparison to other LLM conformal methods. Providing a few empirical comparisons against other work in uncertainty quantification and alignment would help demonstrate the advantages of the proposed method.

**Questions:**

- The term "unit" should be explicitly defined before using throughout the paper
- How is alignment with human experts typically measured? How to generate ground-truth alignment scores? More discussion should be included
- "can be gravy" --> "can be grave"
- In line 114, what, specifically, is the "selection bias"?
-  In line 119, please cite the "Two works addressing the alignment of foundation models with CP ideas have been briefly discussed in Section 1."
- In line 131, what is "mild" about the exchangeability condition and how does it differ from the conventional conformal prediction assumption?
- In line 137, please provide the citations for the "commonly-used alignment criteria:
- How would this framework handle distribution shifts or in-context learning?
- Can a reward model such as in RLHF be used as the alignment predictor? How does alignment relate to human preferences?
- Did you try varying $M$ outputs during the training of the alignment predictor?

**Limitations:**

* How to select an appropriate alignment function? More discussion should be provided as it is the core component of the method
* How to generalize beyond calibrating single tasks to multiple tasks? A user generally queries a foundational model for many different types of tasks, not only a single task.
* How to handle dependence between inputs e.g. a chatbot's output depends on prior context and history. How would this be incorporated?
* The paper would benefit from a dedicated limitations section.

---

> ### Author Rebuttal · Authors · 2024-08-07
>
> Thank you for appreciating the importance of this topic, the writing quality, and the flexibility and versatility of our framework! We respond to your insightful questions in the following.
>
> - **Q1: Technical novelty**. Thank you for the question! While this work applies the conformal selection framework, this presents unique challenges in (1) justifying the criterion for reliability in LLMs/foundation models, (2) formulating the problem into a form that can be tackled by conformal selection, and (3) implementation techniques such as feature selection, alignment prediction training. Please refer to our response to [Q1 of R1] for a more detailed discussion.
>
> - **Q2: Comparison with other LLM conformal methods**.
>     - Thanks for this insightful question. It is worth differentiating the goals of existing LLM conformal papers. We mainly focus on two examples: [CLM, 3] and [conformal factuality, 4].
>         - [3] focuses on the generating process of LLM. In other words, CLM utilizes conformal risk control to determine the stop time for answer generating and for each question. It produces a set of answers that may contain the correct answer (under certain metrics) with a high probability.
>         - [4] focuses on modifying the generated answers to achieve factuality on average.
>         - In comparison, our approach focuses on the “black-box” scenario where we already have access to the inputs and outputs of foundation models, in which our task is to identify a subset of outputs that align with human-verified answers and can be used directly.
>
>     - The major distinction between our method and these existing works is the type of guarantee we pursue. This is related to the following question: how can we use the outputs of these algorithms in practice? In short,
>         (a) For CLM, it is unclear how to pick one answer from the multiple answers in a way that does not hinder validity.
>         (b) Conformal factuality makes the original outputs more vague, which can hinder downstream use.
>
>         - Note that point (b) has recently been noticed (after the submission of our paper) in the literature [12], hence we refer to certain recent works that demonstrate this point for simplicity. In contrast, we make no modification to the model outputs but instead filter out undesirable ones; thus, the quality of good outputs is not affected.
>
>         - In addition, to demonstrate (a), we conducted additional experiments for more clear comparison. With the TriviaQA dataset and the base model LLaMA-2-13B-chat, we follow the procedure in [3] and have the following examples:
>             - Example 1: Question: In tennis, losing two sets 6-0 is known as a double what? True answer: Bagel. Generated set: {‘bagel’}.
>             - Example 2: Question: What was the name of the brothel in The Best Little Whorehouse in Texas? True answer: Chicken ranch. Generated set: {'miss monas', 'chicken ranch', 'miss mona’s', 'miss monas bordello'}
>             As we can see in Example 1, the generated set is very informative and can be used directly. However, although the generated set in the second example has a high probability of covering the true answer, it contains very different answers and is potentially confusing in practice. In other words, whether “Chicken ranch” or “Miss Monas” is correct is unclear to the user.
>             In addition, as is shown in the attached Figure 2 (left panel), we present the proportion that the size of generated size is k for different values of k, which shows that there are more than 40% of sets having no less than 4 distinct items. In addition, Figure 2 (right panel) shows that in nearly 30% of the questions, the admissible answers (with rougeL >= 0.3) take up less than 20% of the uncertain set, which poses a challenge in the direct deployment of answers in the uncertain set.
>
> - **Q3: Definition of “unit”**.
> Thank you for raising this issue and we apologize for potential confusion in the usage of “unit”. We use each “unit” to broadly refer to a data point (a question in Q&A task, and a radiology instance in report generation). In our revision, we have clarified the definition of a “unit” in the introduction.
> - **Q4: Measurement of alignment**. Thank you for the thoughtful question! Indeed, an important problem is the acquisition of reference information to compute the true alignment score.  In general, the alignment criterion depends on the context and can be freely chosen by the practitioner. Currently, for consistency, our experiments with Q&A and radiology report generation follow existing definitions of “acceptable/reliable” outputs, such as factuality (correctness of answers) and consistency with human reports [3,4]. These alignment scores can be computed with existing datasets, yet we expect other notions of alignment that may suit other application contexts. If one is to deploy our method in a new context, they can use common criteria or let humans rate the alignment if feasible. We will add a discussion on this point in our updated manuscript.
> - **Q5: Wording**. Thank you for the detailed suggestions! We’ll modify the wording in our updated draft. Please consider all of them fixed!
>
> - **Q6: Exchangeability**. Our exchangeability condition needs the calibration and test data to be exchangeable, which is identical with the conventional conformal prediction assumption.
> - **Q7: Citations**. We have added the citations of [3,4] in line 119 and [5,10,11] in line 137.
>
> Please see our response to the remaining questions in the Official Comment.
>
> We sincerely hope these comments help clarify your questions and improve your evaluation of our work. Please do not hesitate to let us know if you have any other comments/questions!
>
> **Reference** Please see references in the global rebuttal.

---

> ### Author Response · Authors · 2024-08-07
> **Response to Reviewer SzCp**
>
> - **Q8: Distribution shift and in-context learning**. Thank you for the thoughtful question! There are recent interests in handling distribution shifts in conformal selection, and theoretically, it is feasible to adjust for covariate shift [9] with a proper estimator for the density ratio. This leads to a new definition of p-values and selection procedure. For in-context learning, as long as a certain exchangeability structure is preserved (for example, when the in-context prompts are generated in an exchangeable way), our method is still able to deliver FDR control.
>
> - **Q9: RLHF**. Thank you for the thoughtful comments!
>     - From our limited understanding of RLHF, it focuses on improving the foundation model itself, where the reward function is mainly a stepping stone for better training. In theory, it is completely feasible to use the reward model in our framework since our method applies to any criterion and can use any predictor. However, to incorporate RLHF in our setting, it might not be clear what the “ground-truth” labels are.
>     - Our definition of alignment is remotely related, but differs from the usual definition of human preference. Usually, human preference uses pairwise comparison to overcome the difficulty of a numerical measure of quality. In contrast, we follow the widely adopted notion of “acceptable/reliable” outputs. An interesting future direction would be to allow for using preference outcomes in our framework, but we expect that it would need a different formulation of the problem.
>
> - **Q10: Varying M**. Thanks for this insightful question! The choice of M, the number of generations for each question/CXR image, is very important in estimating the uncertainty/confidence scores. More concretely, a large M could produce better-calibrated scores (e.g. the score is more informative to predict the alignment with human evaluations). The effect of M has been explored in [10] (Table 7, page 22), where they found that the uncertainty/confidence scores with M=3 already outperform white-box scores (e.g. likelihood) and semantic entropy. In addition, the results with M=10 are very close to those with M=20 and the benefit in increasing M is minor when M >= 20. Thus, we opt to use $M=20$ for the best performance in our work. In our revision, we will add more ablation studies with smaller and larger values of $M$, and we expect that the findings should be consistent with those in [10].
>
> - **Q11: Multi-task**. The multi-task setting is indeed an important setting in  LLM. From the theoretical perspective, the current framework can be directly generalized to the following two scenarios: (1) each unit has the same set of tasks or (2) the set of tasks for all units is generated in an exchangeable manner. In both settings, directly applying our framework leads to FDR control. If the tasks differ yet units are exchangeable within each task (which is a very mild condition), one can apply our framework within each task, similar to Mondrian Conformal Prediction, and this leads to FDR control conditional on each task. From the empirical perspective, it would be an interesting open question to design an efficient predictor g(X) to measure the alignment of a multivariate target. In this case, we may need to incorporate information about the task into the features as well.
>
> - **Q12: Dependency**. Thank you for the insightful comment! The only requirement of our method is exchangeability among calibration and test units, so if the inputs are dependent but exchangeable, our theory still applies. In the chatbot example, if we view each conversation as a unit (and aim at selecting conversations), then we still have exchangeability across units. We acknowledge that the exchangeability structure might be broken if the units correspond to former and latter parts of a series of discussions. This issue could be understood as (i) temporal shift in an online setting, or (ii) covariate shift where the covariate is the prior context and history. For each perspective, there are potential remedies. First, if we view it as an online selection problem, we might use ideas in online multiple testing to hedge the FDR budget as data accumulates. Second, if we view it as a covariate shift problem, then conformal selection can be extended to this setting [9], and the problem boils down to estimating a weight function that adjusts for the covariate shift. Both perspectives can bring new techniques to be incorporated into our current framework, and we will add a discussion about this point in our updated draft.
>
> - **Q13: Limitation section**. Thank you for the constructive suggestion! In our updated manuscript, we will add discussions on the following points in a new limitation section:
>     - More broad definition of alignment and prediction such as RLHF
>     - New ideas for handling distribution shift and in-context learning
>     - Evaluating our method with more powerful foundation models and larger-scale datasets

---

> > ### Comment · Reviewer_SzCp · 2024-08-12
> >
> > I thank the author for their detailed response and for addressing my comments. I will raise my score.

---

### Official Review · Reviewer_Ctqb · 2024-07-20

**Soundness:** 2
**Presentation:** 2
**Contribution:** 2
**Rating:** 4
**Confidence:** 4

**Summary:**

This paper introduces Conformal Alignment, a framework designed to ensure that outputs from foundation models align with human values before deployment in high-stakes tasks, such as radiology report generation. By training an alignment predictor with reference data, the framework guarantees that a user-specified fraction of selected units meet the alignment criterion. Demonstrated through applications in question answering and radiology report generation, this method effectively identifies trustworthy outputs with lightweight training over a moderate amount of reference data.

**Strengths:**

1.	This paper studied an important and relevant research problem about when to trust the output of foundation models before deploying them for downstream tasks.
2.	This proposed solution is demonstrated by two applications: question and answering and radiology report generation.
3.	The results show an achievement of strict error control through post-hoc adjustment/selection.

**Weaknesses:**

The paper's technical contribution seems modest, with some ambiguity in distinguishing its novelty from [17], a potentially overstated claim regarding model alignment with human values, and limited comparisons to relevant works.

1.	The technical contribution of the paper appears incremental. As claimed, the main achievement lies in its strict error control through post-hoc adjustment/selection, rather than developing new training strategies to improve specific alignment metrics of the model. However, its theoretical foundation closely resembles that of [17]. For instance, Theorem 3.1 is a modification of a theorem in [17], and Algorithm 1 is also very similar to an algorithm in [17]. The paper lacks clarification on what constitutes the new contributions compared to [17]. It seems that this work merely replaces the monotone nonconformity score V in [17] with a predicted alignment score. Is that right? Additionally, the paper is not well-written, which hampers the understanding of its contributions.

2.	The proposed Conformal Alignment claims to ensure that outputs from foundation models align with human values. However, the evaluation on the radiology report generation task does not support this claim. The alignment scores used to train the score predictor are based on the CheXbert score for 14-class classification, which does not strictly align with human ratings ([R1] Figure 3). The authors should consider using datasets with genuine human ratings, like those in [R1], for verification. Otherwise, this claim seems overstated.
[R1] “Evaluating progress in automatic chest x-ray radiology report generation”. Patterns, 2023


3.	The proposed method was not compared to any relevant works like [16] and [17], except for a heuristic baseline in Figure 4. Without such comparisons, it is difficult to position the proposed method within the existing literature.

**Questions:**

1.	Could you specify the new contributions of your work compared to [17]? It seems Theorem 3.1 and Algorithm 1 closely resemble those in [17], with the primary change being the replacement of the monotone nonconformity score V with an alignment score predictor. Can you elaborate on any additional novel aspects?

2.	Could you please explain the importance and implications of Proposition 3.3 in your framework?

3.	Consider evaluating your method on datasets with actual human ratings to better validate the claim that your framework ensures alignment with human values.

4.	Could you please extend the experimental comparisons to include other relevant methods, such as those in [16] and [17]?

5.	This paper suggested |D|=500. However, if genuine human values are used to train the alignment predictor, this number of human annotations is not small for radiology report generation.  How do you plan to address this challenge?

6.	In Conclusion, it is mentioned that the proposed solution is more relevant to downstream tasks than existing ideas. Could you please elaborate this?

**Limitations:**

Not exactly although some future works were mentioned

---

> ### Author Rebuttal · Authors · 2024-08-07
>
> Thank you for recognizing the importance and relevance of the research question, the applicability and rigor of our framework!
> - **Q1: Comparison with related works**. We would like to clarify that the emphasis of the current work is the novel instantiation of conformal selection for reliable use of foundation model outputs, achieving a practically meaningful guarantee that has not been studied in this thread. While we do not aim to improve the foundation model itself, there are unique challenges in instantiating conformal selection for the current setting.
> In the following, we compare our work with (1) conformal selection [8, 9], and (2) other works that apply uncertainty quantification to LLMs (such as [3,4]).
>     - Compared with conformal selection, we formulate the current problem into a form that can be tackled by conformal selection, and propose practical techniques to make the application useful. More specifically, the major challenge lies in justifying the criteria, formulating the problem, adapting existing techniques to the current setting, and implementing the complete pipeline for practical use. Please see our response to [R1] for a more detailed explanation of these challenges and contributions.
>     - Compared with other works on conformal prediction and LLMs, the major distinction lies in what criterion we aim to achieve (and subsequently, how we achieve them). We argue selecting reliable outputs (similar to the “prediction with abstention” idea) with FDR control, which ensures that most of the deployed outputs (such as the radiology reports that are handed over to doctors) are indeed reliable and useful. This contrasts our work with the works that apply conformal prediction to standard classification/regression tasks.
>     - Particularly in the context of LLM/foundation models, our proposal differs substantially from conformal language modeling [3] that generates multiple outputs for a unit, ensuring *at least* one of the output is aligned, and conformal factuality [4] that edits the original outputs to make it less specific/informative, so as to achieve guarantees for a single unit. More comparisons with the works connecting conformal prediction and LLM/foundation models can be found in Section 2.1 of our manuscript.
>     We also appreciate the comment on the quality of paper writing. We have made efforts to clarify multiple points other reviewers have suggested. In addition, we would be more than happy to address any specific comments the reviewer may have.
> - **Q2: Evaluation with real human ratings**. Thank you for pointing out the potential discrepancy between CheXbert score and the real human rating! In our manuscript, the cheXbert score can be viewed as a proxy for the human rating. Our framework can similarly be applied to the alignment scores given by human evaluation; however, due to limited resources of such data, we shall leave this for future work. (We note that the data from [R1] are human annotations that compare generated reports from a given model against radiology reports in the CXR dataset we used. The total sample size is only 50; while it is possible to train a prediction model with such data (in practice, we only need the training and calibration data to be labeled), it is difficult for us to split out another fold of test data to complete the evaluation process with this dataset. Also, it is difficult to increase the available sample size due to its human annotation nature. Given this limitation, we were unable to directly use the resources in [R1] to validate our framework.) The use of CheXbert score is mainly to demonstrate the general applicability of our framework to any alignment criterion; thus, switching to real human ratings does not affect the validity., In the revision, we have modified our claim, acknowledging that there might be the discrepancy between the proxy and scores by real humans, and discuss human rating as an important future direction.
> - **Q3: Extending the experimental comparison to Jin and Candès [8, 9]**. As mentioned earlier, our current work is an instantiation of the conformal selection framework in the context of LLM alignment; that is, the method being implemented in our experiments is essentially the (adaptation of) the method proposed in [8]. Thus, we may not be able to compare with them in experiments. (However, we do note that this adaptation presents its unique challenges; for a detailed discussion on such challenges, please refer to our answer to [Q1 of Reviewer1]). We would be happy to compare with other candidate methods if you are referring to other methods.
>
> Please see the response to the remaining questions in the Official Comment. We sincerely hope these comments help clarify your questions and improve your evaluation of our work. Please do not hesitate to let us know if you have any other comments/questions!
>
> **[R1]** “Evaluating progress in automatic chest x-ray radiology report generation”. Patterns, 2023

---

> ### Author Response · Authors · 2024-08-07
> **Response to Reviewer Ctqb**
>
> - **Q4: Sample size/Power**. Thank you for this thoughtful question! We totally agree that the sample size is a critical consideration in practice, and that is why we discussed this issue via (i) theoretical analysis and (ii) numerical experiments. In the following, we first summarize the empirical and theoretical results we establish, and discuss further plans to address this challenge.
>     1. **Current results**. First, our experimental results show that, with reasonably powerful models (which are smaller than state-of-the-art, ultra-large models), we can achieve satisfactory power with a few hundred samples. When the number of such data is limited, our method honestly reflects the amount of confidence we have in the alignment status of the samples (if the power is low), as opposed to blindly deploying the outputs without guarantees. Second, as suggested by Proposition 3.3, the two key driving factors of the power are the predictive power of the foundation model and the quality of the alignment score predictor $g(\cdot)$. We expect that one could boost the power of the whole pipeline by improving the quality of the foundation model or working with more powerful foundation models, if this is feasible.
>     2. **Plan for addressing the challenge**. We totally agree that one important future direction is to develop methods that could improve the performance of the selection procedure when the training sample size is limited. We outline a few ideas in the following.
>         - *Improve base foundation model*. Since the power of selection improves with the power of models, we anticipate that if the foundation model itself is more powerful, fewer samples would be needed for training.
>         - *Find more informative features*. Another important factor is whether the features $X$ obtained from the foundation model are informative/predictive of the unknown alignment score. While we currently use existing ideas in the literature for consistency, it will be helpful to identify better features.
>         - *Improve data usage*. The current framework splits labeled data into calibration and training fold, which reduces the sample size in alignment predictor training. However, it might be possible to leverage ideas similar to full conformal prediction to make sure (almost) all labeled data can be used in training. This would greatly increase the computation complexity, yet might be worthwhile if the labeled data are quite limited.
>         - *Improve method development*. We may also extend conformal selection to allow for more flexible and powerful selection procedures. For example, we may use the covariates in the calibration data to increase the effective training sample size while preserving the rigorous error control of the framework. We leave this for future methodological work.
>
>     In summary, we have added a discussion on this problem and outline possible ideas for improvement in our revision.
>
> - **Q5: elaborating on relevance**. The statement is made in comparison with other works that try to connect conformal inference with LLM/foundation models. The main point is how practitioners might use the outputs of these algorithms and what guarantees make sense. While these existing works provide uncertain quantification (similar to prediction sets in conformal prediction), the guarantees they provide usually do not account for downstream implications, i.e., how to use them in subsequent tasks, which is the focus of this paper. Among the representative ideas in the literature [4] applies conformal prediction to obtain a less specific statement than the original generated response that is guaranteed to be correct (but possibly less informative) — however, this might make originally powerful outputs more vague, which hinders downstream use. On the other hand, [3] samples multiple outputs until one of them is reliable with high probability. However, practitioners still have to decide which one is the “good” output, among multiple of them; if one only uses [3] when the set of outputs is a singleton, then the adopted ones suffer from high error (please refer to our answer to Q2 of R5 for new numerical evidence, which is also in Figure 2 of the attached pdf).
>     In contrast, by FDR control, we ensure that practitioners can directly use the outputs selected by our method in downstream tasks, because most of them (such as 90%) are correct. This is without any modification to the original output, which preserves the sharpness of the foundation models.
>
> We sincerely hope these comments help clarify your questions and improve your evaluation of our work. Please do not hesitate to let us know if you have any other comments/questions!
>
> **Reference** Please see references in the global rebuttal.

---

> > ### Comment · Reviewer_Ctqb · 2024-08-13
> >
> > Dear authors,
> >
> > Thank you for your feedback. Indeed, my major concern is the resemblance of this paper to [17]. Could you please clarify that? "For instance, Theorem 3.1 is a modification of a theorem in [17], and Algorithm 1 is also very similar to an algorithm in [17]. The paper lacks clarification on what constitutes the new contributions compared to [17]. It seems that this work merely replaces the monotone nonconformity score V in [17] with a predicted alignment score. Is that right?" Thanks.

---

> > > ### Comment · Reviewer_Ctqb · 2024-08-13
> > >
> > > If this concern (resemblance to [17]) cannot be cleared, I have to maintain my original score.

---

> > > > ### Author Response · Authors · 2024-08-13
> > > >
> > > > Thank you for the feedback. Indeed, we apply conformal selection to LLMs (similar to related works that apply conformal prediction to LLMs e.g., [3,4]). However, as we have mentioned in our previous rebuttal, the emphasis of the current work is the novel instantiation of conformal selection for reliable use of foundation model outputs, which is non-trivial. Compared with conformal selection, the major challenge lies in justifying the criteria, formulating the problem, adapting existing techniques to the current setting, and implementing the complete pipeline for practical use. We elaborate on the challenges in the following.
> > > >
> > > > - The first challenge is justifying what criterion is practically useful. We argue that selecting reliable outputs (similar to the “prediction with abstention” idea) with FDR control is a practically reasonable criterion since it ensures that most of the deployed outputs (such as the radiology reports that are handed over to doctors) are indeed reliable and useful. This contrasts our work with the works that apply conformal prediction to standard classification/regression tasks. Particularly in the context of LLM/foundation models, our proposal differs substantially from conformal language modeling [3] that generates multiple outputs for a unit, ensuring *at least* one of the outputs is aligned, and conformal factuality [4] that edits the original outputs to make it less specific/informative, so as to achieve guarantees on the correctness of the output for a single unit. More comparisons with the works connecting conformal prediction and LLM/foundation models can be found in Section 2.1 of the manuscript.
> > > >
> > > > - The second challenge is formulating the alignment problem as a selective prediction problem that can be tackled with conformal selection, and adapting conformal selection to the current setting. Note that while conformal selection is an established method, its current applications in job hiring and drug discovery apply to numerical outputs and predictions, rather than aligning foundation model outputs (which do not come with a clear numerical output). To make conformal selection applicable to our purpose, we need to formulate alignment as a numerical indicator that could be inferred with reference information, and then decide how to train a prediction model for the alignment score, which can therefore be used in conjunction with the conformal selection framework. Indeed, how conformal prediction ideas may be used to calibrate LLMs/foundation models (beyond traditional numerical outputs) is a nontrivial problem that has attracted considerable recent interest; in this sense, we contribute to discussion on this important issue.
> > > >
> > > > - With the general recipe, the third important challenge and contribution is the practical instantiation of the pipeline, which features to use, what model to train for alignment, among other considerations. In this regard, we conduct extensive numerical experiments, finding that using a lightweight prediction model such as logistic regression or random forests with popular heuristic (uncalibrated) uncertainty measures is often sufficient for identifying reliable outputs. This provides a concrete and easy-to-use pipeline for practical use. Indeed, even beyond the current setting of selecting reliable outputs, we view the choice of features and prediction models as contributions that may be helpful for predicting the reliability of outputs for other purposes.
> > > >
> > > > Hope this clarifies the confusion!

---

> ### Comment · Area_Chair_cH34 · 2024-08-12
> **Authors responses**
>
> Dear reviewer,
>
> did authors address your concerns in their rebuttal? I would appreciate if you could review their responses and potentially engage in a discussion with them?
>
> Best,
> your AC.

---

> ### Comment · Reviewer_Ctqb · 2024-08-14
>
> Dear Authors,
>
> Thank you for your clarification. I truly understand and appreciate your contribution from the application perspective, which is the reason for my original score suggestion. However, the theoretical part (including the algorithm), which is a main component of the proposed work, seems to have a large overlap with the existing work [17].  I could not see how the algorithm in [17] is extended to address the application challenges in this work, which impacts my understanding of the novelty of this paper. Meanwhile, due to commonly very limited genuine human rating data (instead of machine-generated CheXbert scores), the proposed method needs to carefully handle a small sample-size problem. I appreciate the authors' response and suggest incorporating this into the limitation of the current approach. Due to the above reasons, I keep my original score.

---

> > ### Author Response · Authors · 2024-08-14
> > **Follow-up comment**
> >
> > Thank you so much for the follow-up!
> >
> > For your concern on evaluation metrics, it's worth mentioning that in $\mathcal{D}_{\rm train}$, both generated and human-written reports are given and the difference lies in how to define the alignment map $\mathcal{A}$ in practice. As the limitation of human annotations can be inevitable in some scenarios, we currently follow the practice in [1], where $\mathcal{A}$ is a proxy for human rating based on the CheXbert score. As is shown in the reference [2] provided by the reviewer, existing metrics can be imperfect in terms of the alignment with human ratings, thus it is a very interesting open question (but is also of independent interest) in the area. We will discuss this in our updated draft!
> >
> > [1] Quach, Victor, et al. "Conformal language modeling." arXiv preprint arXiv:2306.10193 (2023).
> >
> > [2] Yu, Feiyang, Mark Endo, Rayan Krishnan, Ian Pan, Andy Tsai, Eduardo Pontes Reis, Eduardo Kaiser Ururahy Nunes Fonseca et al. "Evaluating progress in automatic chest x-ray radiology report generation." Patterns 4, no. 9 (2023).

---

### Official Review · Reviewer_FNNT · 2024-07-27

**Soundness:** 4
**Presentation:** 3
**Contribution:** 3
**Rating:** 6
**Confidence:** 4

**Summary:**

This paper proposes a procedure similar to conformal prediction to select deployable units and control the FDR. Asymptotic result is provided to justify the correctness of the proposed algorithm. Real data experiments in QA and medical tasks are also provided to demonstrate the effectiveness of the proposed method.

**Strengths:**

The mathematical formulation, theoretical justification, and basic experimental setups are all clearly and well written. The theoretical results are appreciated because they can provide more rigorous statistical guarantees on the proposed method.

**Weaknesses:**

While the theoretical results are appreciated, the major issue of this paper lies in its writing:

(1) The word "alignment" in this paper is different from the very commonly used "alignment" in which LLMs are tuned to refuse to answer harmful questions. This wording issue can cause significant confusion when readers reading this paper. Please consider replacing the word with other choices to better describe the task performed in this paper, e.g., calibration?

(2) The authors are suggested to better clarify their contribution compared to existing literature. In terms of the methodology, how is the algorithm different from the things in [5,16,17] in the reference list? For technical contribution, to set an example, in line 167, it is mentioned that "The proof of Theorem 3.1 is adapted from [17], and we include it in Appendix A...". Please consider summarize the adaptation and challenges when deriving Theorem 3.1.

(3) While the title of this paper mentions "foundation model", it is not clear how the proposed algorithm is related to foundation models: it seems that the proposed method can also be used in other small-scale models. Alternatively, is there any other better methods for small-scale models than the proposed method which cannot be used for foundation models?

(4) Please define "power" in line 172.

(5) Line 255 "More powerful model enables more powerful selection": could you provide insights or interpretation on this observation?

(6) Figure 5: XGBoost provides the functionality of computing the importance score for the features used in the regression. In addition to the individual factors considered in logistic regression in Figure 5, could the authors provide more results on how important each features is? The correlation among features may also makes the actual importance of the features when regression on them together.

(7) Non-asymptotic result: Although Proposition 3.3 provides the asymptotic result on the power of the method, since the experiment includes "How many high-quality samples are needed?", the authors are also suggested to provide a discussion on the non-asymptotic results.

**Questions:**

My rate is 4 for now because of the many issues in this paper. I still value the theoretical justifications and would like to raise my score if the authors are willing to clarify them.

**Limitations:**

No limitations.

---

> ### Author Rebuttal · Authors · 2024-08-07
>
> Thank you for appreciating the theoretical rigor and the writing of our paper! Please see below our detailed responses.
> - **Q1: Use of the term “alignment”**. Thank you for pointing out the potential confusion! In fact, our proposed method can also be used for “alignment” in the strict sense: if we define our alignment score to reflect whether an output is harmful, then applying our method is essentially tuning the LLM to refrain from deploying harmful responses with theoretical guarantees. In this sense, we are using the word “alignment” a bit loosely in our paper, since we allow “alignment” to correspond to other criteria depending on the context. To avoid potential confusion, we have included a disclaimer in the introduction explaining the difference in our updated manuscript. We welcome more discussion on the choice of terminology if you have further concerns!
> - **Q2: Clarifying Contributions**. We would like to clarify that the emphasis of the current work is the novel instantiation of conformal selection for reliable use of foundation model outputs. Compared with conformal selection, the major challenge lies in justifying the criteria, formulating the problem, adapting existing techniques to the current setting, and implementing the complete pipeline for practical use. We elaborate on the challenges in the following.
>
>     - The first challenge is justifying what criterion is practically useful. We argue that selecting reliable outputs (similar to the “prediction with abstention” idea) with FDR control is a practically reasonable criterion, since it ensures that most of the deployed outputs (such as the radiology reports that are handed over to doctors) are indeed reliable and useful. This contrasts our work with the works that apply conformal prediction to standard classification/regression tasks. Particularly in the context of LLM/foundation models, our proposal differs substantially from conformal language modeling [3] that generates multiple outputs for a unit, ensuring *at least* one of the outputs is aligned, and conformal factuality [4] that edits the original outputs to make it less specific/informative, so as to achieve guarantees on the correctness of the output for a single unit. More comparisons with the works connecting conformal prediction and LLM/foundation models can be found in Section 2.1 of the manuscript.
>
>     - The second challenge is formulating the alignment problem as a selective prediction problem that can be tackled with conformal selection, and adapting conformal selection to the current setting. Note that while conformal selection is an established method, its current applications in job hiring and drug discovery apply to numerical outputs and predictions, rather than aligning foundation model outputs (which do not come with a clear numerical output). To make conformal selection applicable to our purpose, we need to formulate alignment as a numerical indicator that could be inferred with reference information, and then decide how to train a prediction model for the alignment score, which can therefore be used in conjunction with the conformal selection framework. Indeed, how conformal prediction ideas may be used to calibrate LLMs/foundation models (beyond traditional numerical outputs) is a nontrivial problem that has attracted considerable recent interest; in this sense, we contribute to the discussion on this important issue.
>
>     - With the general recipe, the third important challenge and contribution is the practical instantiation of the pipeline, which features to use, what model to train for alignment, among other considerations. In this regard, we conduct extensive numerical experiments, finding that using a lightweight prediction model such as logistic regression or random forests with popular heuristic (uncalibrated) uncertainty measures is often sufficient for identifying reliable outputs. This provides a concrete and easy-to-use pipeline for practical use. Indeed, even beyond the current setting of selecting reliable outputs, we view the choice of features and prediction models as contributions that may be helpful for predicting the reliability of outputs for other purposes.
>
>
> - **Q3: Small model/foundation model**. You are absolutely correct that our method is not limited to foundation models, and can be generally applicable to problems of any size. The paper is titled “foundation model” because this is the primary question we find important and aim to address in this paper; in particular, substantial work is devoted to the instantiation of our framework for the alignment of foundation model outputs, e.g., the choice of criterion, building models for predicting the alignment score, among others.
>
>    For smaller base models, we might be able to consider more complicated models for training $g(\cdot)$; for example, we may include all raw features for training $g$.This technique might be prohibitively expensive for foundation models.
>
> - **Q4: Definition of power**. Thank you for the suggestion! The definition of power was given in the equation between lines 96 and 97. To improve clarity, we have added a reminder of the definition of power in Proposition 3.3 in the revision.
>
> Please see our response to the remaining question in the Official Comment.
>
> We sincerely hope these comments help clarify your questions and improve your evaluation of our work. Please do not hesitate to let us know if you have any other comments/questions!

---

> ### Author Response · Authors · 2024-08-07
> **Response to Reviewer FNNT**
>
> - **Q5: More powerful model enables more powerful selection**. Thank you for raising this question for discussion. This statement can be illustrated from both empirical and theoretical perspectives. (1) In practice, the alignment scores are correlated with the model likelihood,  input uncertainty, and output confidence produced by the model (line 209), so with more capable models, the alignment scores could be better calibrated. Taking Q&A as an example, the alignment scores with LLaMA-13b could be in line with the true correctness of the output more faithfully, thus the selection power is higher as is shown in Figure 3. (2) From the theoretical perspective, when the alignment score predictor g(x) is more powerful, e.g. g(x) \approx A, by proposition 3.3, the asymptotic power can be maximized. In addition, [8] shows via Neyman-Pearson Lemma that the optimal predictor should be monotonic in P(A>c|X).
>
> - **Q6: Feature importance**. Thank you for the thoughtful question! We would like to first clarify that the purpose of the ablation study on feature importance is mainly to provide information on the choice/usefulness of features that might be helpful for future related tasks, which is orthogonal to our main point of validity.
>
>     Following your suggestions, we used the XGBoost functionality to compute the importance scores (please see Figure 1(d) in the attached pdf). We also computed the Shapley value of each feature to measure their importance based on the prediction models we use in the manuscript. As the Shapley values are model-agnostic, we present results for three choices of g(X) with different base classifiers, including logistic regression, random forests, and XGBoostRF, in Figure 1(a)(b)(c). Although the exact order of features varies in four plots, there are features that have consistent dominating effects in all figures, e.g. Deg(J) and EigV(J) based on the Jaccard similarity. In addition, NumSets as the feature in alignment score predictors tends to exhibit an effect close to zero in all four figures. We should note that the aforementioned findings are also revealed in Figure 5 in the paper, where predicted scores with EigV(J), Deg(J), and Ecc(J) have higher power and that with NumSets is much less powerful. We thank the reviewer again for raising this question for more comparison of feature importance in the alignment score predictor.
>
> - **Q7: non-asymptotic power**. Thank you for the suggestion! We will be adding a parallel non-asymptotic result on power in the revised manuscript. To derive the non-asymtotic result, we leverage the uniform non-asymptotic concentration inequalities, obtaining an error bound on the scale of $1/\sqrt{n_{\text{calib}}}$.
>
> We sincerely hope these comments help clarify your questions and improve your evaluation of our work. Please do not hesitate to let us know if you have any other comments/questions!
>
> **Reference** Please see references in the global rebuttal.

---

> > ### Comment · Reviewer_FNNT · 2024-08-07
> >
> > I appreciate the authors in addressing my comments. It is now clearer to me the contribution of this paper, so I raised my score to 6.

---

### Official Review · Reviewer_w89M · 2024-07-27

**Soundness:** 4
**Presentation:** 4
**Contribution:** 4
**Rating:** 5
**Confidence:** 4

**Summary:**

This work applies recent work on conformal outlier detection [1] in the context of LLM alignment. The authors assume that one, given input $X$ and output $Y$ of an LLM, has access to an alignment scorer $A$ that can then produce an alignment score of $A(X, Y)$. They treat the null hypothesis of being unaligned as being $H_0: A(X, Y) \leq c$, i.e., the alignment score being smaller than a threshold $c$. They describe the typical score function as $\widehat{A}$, and the resulting conformal score being $\widehat{A}(X)$ for a data point. Using the scores, they construct a conformal p-value that is superuniform when $A(X, Y) \leq c$ and apply the Benajmini-Hochberg procedure on the conformal p-values to get FDR control.

**Strengths:**

This paper is theoretically rigorous and applies the framework of conformalized selection to "alignment" of LLMs, which is an application that is not directly explored in previous papers on conforamlized selection. The experiments show that with a small amount of data, one can derive powerful procedures with FDR control when using off the shelf open source models, and that's nice. It is interesting that a relatively "weak" (computationally) scoring model can still produce good results.

**Weaknesses:**

I think the paper, in some sense, is just an extra series of experiments for conformalized selection --- we already know it works well for many deep learning tasks from previous papers, and the experimental results are not particularly surprising, since it continues that trend. Many other conformal papers in the LLM space try to tackle the problem that's unique to LLMs, i.e., factuality, and try to investigate the fact that it is not already a pre-defined, structured task, and aim to apply conformal in a meaningful way. Thus, I think this paper does lack some novelty in its methods.

**Questions:**

- What is $q$ in Prop. 3.3 --- is it some constant less than $\alpha$?
- What is the cost in power in directly estimating the asymptotic cutoff (and getting a PAC bound) vs. the conformal procedure?

**Limitations:**

Yes, the limitations are sufficiently addressed.

---

> ### Author Rebuttal · Authors · 2024-08-07
>
> Thank you for appreciating the rigor, power, and versatility of our framework! We address your comments in the following.
>
> - **Q1: Novelty**. Thank you for raising this question! We would like to argue that the motivation of this paper is exactly to address the unique challenges of LLMs—the reliability in adopting their outputs—with conformal selection techniques. In the process of applying conformal selection to this problem, there are nontrivial challenges including problem formulation, criteria justification, and practical instantiation of the framework (which is at least as important as the challenges other works address in applying conformal prediction to more traditional classification/regression tasks). We elaborate on the challenges and comparisons with other CP-LLM works below.
>
>
>
>     - **Unique challenge of LLM and “pre-defined tasks”**.
>         - We would like first to argue that the reliability of their outputs (including factuality) is a unique challenge of LLMs, which we aim to address in this work, similar to other related works. Instead of working with “pre-defined” tasks, a lot of our efforts are devoted to finding the appropriate criteria in the context of foundation model deployment, formulating this problem into a tractable form that can be tackled by conformal selection, arguing that it does provide practically meaningful guarantees that have not been previously considered, and developing (practical) techniques that make sure this framework leads to meaningful results. We elaborate on the challenges in the following.
>
>             (1) The first challenge is justifying what criterion is practically useful. We argue that selecting reliable outputs (similar to the “prediction with abstention” idea) with FDR control is a practically reasonable criterion, since it ensures that most of the deployed outputs (such as the radiology reports that are handed over to doctors) are indeed reliable and useful. This contrasts our work with the works that apply conformal prediction to standard classification/regression tasks. Particularly in the context of LLM/foundation models, our proposal differs substantially from conformal language modeling [3] that generates multiple outputs for a unit, ensuring *at least* one of the outputs is aligned, and conformal factuality [4] that edits the original outputs to make it less specific/informative, so as to achieve guarantees on the correctness of the output for a single unit. More comparisons with the works connecting conformal prediction and LLM/foundation models can be found in Section 2.1 of the manuscript.
>
>             (2) The second challenge is formulating the alignment problem as a selective prediction problem that can be tackled with conformal selection, and adapting conformal selection to the current setting. Note that while conformal selection is an established method, its current applications in job hiring and drug discovery apply to numerical outputs and predictions, rather than aligning foundation model outputs (which do not come with a clear numerical output). To make conformal selection applicable to our purpose, we need to formulate alignment as a numerical indicator that could be inferred with reference information, and then decide how to train a prediction model for the alignment score, which can therefore be used in conjunction with the conformal selection framework. Indeed, how conformal prediction ideas may be used to calibrate LLMs/foundation models (beyond traditional numerical outputs) is a nontrivial problem that has attracted considerable recent interest; in this sense, we contribute to the discussion on this important issue.
>
>             (3) With the general recipe, the third important challenge and contribution is the practical instantiation of the pipeline, which features to use, what model to train for alignment, among other considerations. In this regard, we conduct extensive numerical experiments, finding that using a lightweight prediction model such as logistic regression or random forests with popular heuristic (uncalibrated) uncertainty measures is often sufficient for identifying reliable outputs. This provides a concrete and easy-to-use pipeline for practical use. Indeed, even beyond the current setting of selecting reliable outputs, we view the choice of features and prediction models as contributions that may be helpful for predicting the reliability of outputs for other purposes.
>
>
>         - Second, we would like to emphasize that our framework applies to any ``alignment’’ criterion, including the important factuality issue that you have mentioned. The distinction from other works is the criterion we choose and the guarantee we deliver, which we view as an important contribution to the current active discussion in this area. For example, conformal language modeling [3] applies conformal prediction to a single unit, obtaining a less specific statement than the original generated response that is guaranteed to be correct (but possibly less informative); however, to ensure a population-wise guarantee, this may compromise originally informative and reliable outputs (by making them more vague). For another example, the method in [7] makes a selection when the CP set is a singleton; but here there are no theoretical guarantees on the outcome of *selected units*.
>
> Please see the response to the remaining questions in the Official Comment. We sincerely hope these comments help clarify your questions and improve your evaluation of our work. Please do not hesitate to let us know if you have any other comments/questions!

---

> > ### Comment · Reviewer_w89M · 2024-08-12
> >
> > Thank you for the detailed reply. My criticism of novelty is more from its relationship with prior work [8]. The alignment criterion utilized in the experiments in this paper can be rephrased in terms of binary classification (as the alignment labels are simply binary labels in all the experiments), but many experiments of this type have already been carried out in [8]. This paper reformulates many existing "alignment" criteria that have already been established for certain tasks/datasets into binary classification just by picking a threshold to determine "aligned" vs "not aligned", so this is not really novel. By referring to existing work utilizing conformal techniques to LLMs, I was commenting that a key difficulty is actually figuring out what is a meaningful alignment criterion to control --- this paper does not address this difficulty at all.
> >
> > I do think the experimental results are somewhat interesting (particularly the usage of an efficient scorer), so I am sticking with my original score.

---

> ### Author Response · Authors · 2024-08-07
> **Response to Reviewer w89M**
>
> - **Q2: typo in Prop 3.3**.This is a typo and should be $\alpha$ instead. Please consider it fixed!
>
> - **Q3: PAC bound**. If we resort to the pac bounds, we need a high probability upper bound on the FDP that is uniform over the rejection threshold $t$ (since $t$ is chosen in a data-driven manner and the FDR is not monotone in $t$), which amounts to an error of $\frac{c}{\sqrt{n_{\text{calib}}}}$ with a potentially large constant $c$. Thus, if we would like to achieve an FDR below $\alpha$ with high probability, we need to run the BH procedure at nominal level $\alpha - \frac{c}{\sqrt{n_{\text{calib}}}}$.  When $n_{\text{calib}}$ is a few hundred, the gap is on the scale of $0.1$, which may have a huge impact on the power of our method, as manifested in the experiments.
>
> We sincerely hope these comments help clarify your questions and improve your evaluation of our work. Please do not hesitate to let us know if you have any other comments/questions!
>
>
> **Reference** please see references in the global rebuttal.

---

### Official Review · Reviewer_nRXb · 2024-07-29

**Soundness:** 3
**Presentation:** 3
**Contribution:** 2
**Rating:** 6
**Confidence:** 4

**Summary:**

The paper proposes to use conformal prediction to determine when to trust model outputs. To this end the authors apply conformal risk control on across unit instead of focusing on a single unit. This is especially relevant in their X-ray example where the goal is to give a set of trusted documents to the medical professional to review. They show that their adaptation of conformal prediction is also sensible in terms of FDR and power.

**Strengths:**

- The paper is the first to focus on conformal prediction across units compared to previous work that worked mainly on a unit level.
- The paper proposes a new and practical way to use conformal prediction in foundation models for downstream tasks, especially when
- The paper shows on extensive experiments the trade offs between power and FDR control which seems to be better than existing baselines.

**Weaknesses:**

- My main concern with this paper is that, at the end of the day it is just an application of conformal risk control to foundation models. No extensive new theory was needed, but rather just a twist to the setting. If the authors could clearly design the technical challenges faced it would be helpful in making a decision on this paper.
- For the alignment score predictor, can the authors please elaborate on how this score was picked and why this was the best design choice? These seems to be randomly picked and a more systematic justification would be very welcome to better understand if this score can be improved or not
- In practice for a medical professional, what is the target FDR rate? In all experiments the power is quite slow to increase at low FDR rate, however in practice i would have assumed that a medical professional would not want to look through documents where every second one is wrong? This puts into question the motivation as well as practicality of this paper which i would urge the authors to clarify.

i am more than happy to change my score of the above have been addressed.

**Questions:**

see above

**Limitations:**

yes

---

> ### Author Rebuttal · Authors · 2024-08-07
>
> Thank you for the insightful comments, and for appreciating the novelty and practicality of our framework, as well as our experimental results!
>
> - **Q1, Technical challenges**: We would like to clarify that the emphasis of the current work is the novel instantiation of conformal selection (we also note that this is a distinct framework from conformal risk control [1,2]) for reliable use of foundation model outputs. Compared with conformal selection, the major challenge lies in justifying the criteria, formulating the problem, adapting existing techniques to the current setting, and implementing the complete pipeline for practical use. We elaborate on the challenges in the following.
>
>     - The first challenge is justifying what criterion is practically useful. We argue that selecting reliable outputs (similar to the “prediction with abstention” idea) with FDR control is a practically reasonable criterion, since it ensures that most of the deployed outputs (such as the radiology reports that are handed over to doctors) are indeed reliable and useful. This contrasts our work with the works that apply conformal prediction to standard classification/regression tasks. Particularly in the context of LLM/foundation models, our proposal differs substantially from conformal language modeling [3] that generates multiple outputs for a unit, ensuring *at least* one of the outputs is aligned, and conformal factuality [4] that edits the original outputs to make it less specific/informative, so as to achieve guarantees on the correctness of the output for a single unit. More comparisons with the works connecting conformal prediction and LLM/foundation models can be found in Section 2.1 of the manuscript.
>
>     - The second challenge is formulating the alignment problem as a selective prediction problem that can be tackled with conformal selection, and adapting conformal selection to the current setting. Note that while conformal selection is an established method, its current applications in job hiring and drug discovery apply to numerical outputs and predictions, rather than aligning foundation model outputs (which do not come with a clear numerical output). To make conformal selection applicable to our purpose, we need to formulate alignment as a numerical indicator that could be inferred with reference information, and then decide how to train a prediction model for the alignment score, which can therefore be used in conjunction with the conformal selection framework. Indeed, how conformal prediction ideas may be used to calibrate LLMs/foundation models (beyond traditional numerical outputs) is a nontrivial problem that has attracted considerable recent interest; in this sense, we contribute to the discussion on this important issue.
>
>     - With the general recipe, the third important challenge and contribution is the practical instantiation of the pipeline, which features to use, what model to train for alignment, among other considerations. In this regard, we conduct extensive numerical experiments, finding that using a lightweight prediction model such as logistic regression or random forests with popular heuristic (uncalibrated) uncertainty measures is often sufficient for identifying reliable outputs. This provides a concrete and easy-to-use pipeline for practical use. Indeed, even beyond the current setting of selecting reliable outputs, we view the choice of features and prediction models as contributions that may be helpful for predicting the reliability of outputs for other purposes.
>
>
> - **Q2, Choice of alignment score predictor**: Thanks for the insightful question! To avoid any misunderstanding of what you refer to, we discuss the choice of (1) the alignment score $A=\mathcal{A}(f(X),E)$, (2) the alignment predictor $g\colon \mathcal{X}\to [0,1]$, and (3) the features $X$ that are used for prediction.
>
>    - First, the choice of the alignment score $A=\mathcal{A}(f(X),E)$ is context-dependent. In particular, our examples focus on the widely adopted definitions of ``reliability’’ of model outputs in the literature (see the references in the manuscript). For instance, in the X-ray report generation case, we use Chexbert [5] to generate the alignment score, which is widely used in the literature.
>
>    - Second, given features $X$ and the reference, the alignment predictor $g(X)$ should be as accurate as possible for the alignment score $A$ (see the power analysis in Proposition 3.3 that shows how the power of our procedure depends on the accuracy of $g(\cdot)$). Indeed, the optimal $g(X)$ should be monotone in $P(A>c|X)$. In principle, one may use any machine learning model to train this predictor, while we have offered some practical suggestions in our manuscript. Through experiments, we find that relatively lightweight models such as logistic regression and random forests suffice for accurately identifying reliable outputs. This is consistent with other works on selective prediction for LLMs that use tree models [6].
>
>    - Finally, for the choice of features $X$, the general principle is that they should be (1) informative and (2) light-weight (that is, they should be computationally efficient to acquire and train the model $g$ with). In particular,  we choose popular (heuristic) measures of uncertainty in the literature (see the references in the manuscript) as the features, since they shall be informative of the reliability of model outputs; we then train a mapping from the features via lightweight models. As such, the combination of the features is data-driven.
>
> Please see the response to Q3 in the Official Comment. We sincerely hope these comments help clarify your questions and improve your evaluation of our work. Please do not hesitate to let us know if you have any other comments/questions!

---

> ### Author Response · Authors · 2024-08-07
> **Response to Q3 of Reviewer nRXb**
>
> **Q3: FDR target rate**.Thank you for raising this important practical issue! In such high-stakes decisions, we would expect the practitioners to work with a relatively stringent FDR target level; however, we would still like to leave the judgment of an appropriate FDR level to practitioners. Meanwhile, we want to emphasize that the power of the whole pipeline mostly depends on that of the foundation model (please refer to Proposition 3.3 of our manuscript for an explicit characterization of the asymptotic power of our method). Since we enforce error control, its tradeoff with power is inevitable.  Indeed, if the foundation model is unreliable, then it is reasonable that few of its outputs should be deployed in practice (otherwise there will be many *unacceptable* outputs being used, without any alert). In this sense, we provide a rigorous way of distinguishing acceptable outputs against unacceptable ones, adding a layer of protection for the deployment of the foundation model in high-stakes tasks. Such a tradeoff is similar to original conformal prediction: there, if the prediction model (the foundation model in our context) is not accurate, then it necessitates large prediction sets to achieve a prescribed coverage probability. Finally, we anticipate that very large, state-of-the-art models should lead to higher power; while the models we use here are on par with works in the literature [3], we still expect much space for improvement in terms of the quality of the foundation model.
>
> We sincerely hope these comments help clarify your questions and improve your evaluation of our work. Please do not hesitate to let us know if you have any other comments/questions!

---

> > ### Comment · Reviewer_nRXb · 2024-08-08
> > **Response**
> >
> > I thank the authors for their rebuttal.
> > 1. The rebuttal is hard to follow the key points. I had to read over it multiple times to understand most of the idea. At its core teh rebuttal is not arguing that the theory is novel but rather the application. Hence papers likes [Selective conformal inference with false coverage-statement rate control] or [Conformal Prediction Sets with Limited False Positives] are neither cited nor discussed. Can the authors explain if the proposed method in the paper is a simple application of these types of papers? Please be succinct as your previous review already exceeded the 6000 characters and as per rule i would not have to read the additional comments. I believe this sentiment is prelevant amongst most reviewers and hence i would appreciate a clear distinction between the methods that already exist in term of "what can previous methods already do" and "what can the proposed method do that the existing methods cannot"
> > 2. So the authors agree that the selection is indeed data dependent and hence a pure heuristic. Did the authors conduct any ablation study in the case these predictors were badly chosen? i.e. corrupted with noise or have lower signal?
> > 3. i agree with the authors.

---

> > > ### Author Response · Authors · 2024-08-09
> > > **Reply to Reviewer nRXb**
> > >
> > > Thank you for taking the time to read through our rebuttal and follow up on the discussion!
> > >
> > > - Firstly, thank you for pointing out the references [1, 2]; we will add them to the manuscript. other works focus on the **construction of prediction sets** for a single unit, while we focus on using conformal prediction to **select units**. Specifically, [2] constructs a prediction set whose candidate values obey FDR control (which is defined for each unit, while ours is for multiple selected units), and [1] constructs prediction sets for units that are selected by a given process. As such, the techniques we use are also quite distinct.
> > >
> > > - For Q1, we do apply conformal selection to LLMs (similar to related works that apply conformal prediction to LLMs e.g., [3,4]). Our previous rebuttal emphasizes that, not only is the application not straightforward (which involves defining the suitable guarantee and efficient instantiation to maintain power), but the application itself also reveals richer and interesting phenomena for understanding LLMs and uncertainty (e.g. how language model affects the power of selection, what features are informative of reliability, what LLMs’s uncertainty is more predictable).
> > >
> > > - For Q2, we would discuss both the predictors $X$ and the prediction model $g(X)$ in order to minimize the chance of our misunderstanding. To summarize, we offer theoretical justification for the optimal choice of g(X). However, in practice, the model $g$ has to be trained and the choice of features follows the standard in the literature; thus, they are indeed data-dependent.
> > >     - First, the features $X$ we choose are the state-of-the-art measures of uncertainty in the current literature, based on the fact that such measures have been observed to be informative of the reliability of LLM outputs. We totally agree that there is much space for new features, which we leave for future work.
> > >     - Second, the choice of the prediction model $g$ (based on classifiers) is guided by the asymptotic power [proposition 3.3], which is maximized when $g(X)$ is proportional to $P(A > c | X)$ almost surely, and thus we train $g$ to be classification models. This model $g$ is naturally data-dependent to maximize the power of detecting trustworthy outputs.
> > >
> > > - Finally, we totally agree that it is meaningful to conduct additional ablation studies to better understand the performance. In response, we conducted ablation studies when (i) features $X$ are corrupted with noise (**Table 1**), and (ii) the predictor $g(X)$ is corrupted with noise after being trained (**Table 2**).  We can see that in **Table 1**, FDR control is still valid as $g(\tilde X)$ is trained with labeled data although features are corrupted. In **Table 2**, corrupted $\tilde g(X)$ is no longer informative, thus FDR control no longer holds when the noise strength increases. The power in both cases is decreasing with higher noise strength. In summary, we find that model training with informative features is a crucial component in the pipeline.
> > >
> > > We truly hope our comments could clarify your questions and enhance your evaluation of the current paper. Please don’t hesitate to let us know if you have any further questions or concerns!
> > >
> > > **Table 1: features X with entrywise noise N(0, noise strength)**
> > >
> > > noise strength   |   0.0  |  2.5  |  5.0  |  7.5  |  10.0  |  12.5  |  15.0
> > > :---------------:   |  ----- |-----  | ----- | ----- | ------ | ------  | ------
> > > FDR                   |           0.2  |  0.2  |   0.2  | 0.2   |  0.19 |   0.2  |   0.18
> > > Power                |            0.95 |  0.93  |  0.9 |  0.85  |  0.73  |  0.62 |   0.45
> > >
> > > **Table 2: predicted score g(X) with noise N(0, noise strength)**
> > >
> > > noise strength    |  0.0   | 0.2  |  0.4  |  0.6  |  0.8  |  1.0
> > > :----------------:    |-----    |----- | ----- | ----- | ----- | -----
> > > FDR                    | 0.2     | 0.2  | 0.2   | 0.21  | 0.24 |  0.27
> > > Power                 |0.94    | 0.9   | 0.68 |  0.45 |  0.41 |  0.43
> > >
> > >
> > > **References**
> > >
> > > [1] Bao, Yajie, Yuyang Huo, Haojie Ren, and Changliang Zou. "Selective Conformal Inference with FCR Control." arXiv preprint arXiv:2301.00584 (2023).
> > >
> > > [2] Fisch, Adam, Tal Schuster, Tommi Jaakkola, and Regina Barzilay. "Conformal prediction sets with limited false positives." In International Conference on Machine Learning, pp. 6514-6532. PMLR, 2022.
> > >
> > > [3] Quach, Victor, Adam Fisch, Tal Schuster, Adam Yala, Jae Ho Sohn, Tommi S. Jaakkola, and Regina Barzilay. "Conformal language modeling." arXiv preprint arXiv:2306.10193 (2023).
> > >
> > > [4] Mohri, Christopher, and Tatsunori Hashimoto. "Language models with conformal factuality guarantees." arXiv preprint arXiv:2402.10978 (2024).

---

> > > > ### Comment · Reviewer_nRXb · 2024-08-12
> > > > **Thanks for the response**
> > > >
> > > > I am happy with the above response and will increase my score to 6.
> > > > Thanks again for the clarifications and hope that the authors will add them into the final version of their paper.

---

### Author Rebuttal · Authors · 2024-08-07

We thank the reviewers for their valuable feedback and constructive suggestions for improvement. Overall, the reviewers find our work addresses an important problem [R1, R3, R4, R5], is well-written [R1, R5], versatile to broad contexts and criteria [R1, R2, R5], offers rigorous and practically useful guarantees [R1, R3, R4], and offers extensive experimental results [R1, R2, R5].

A few stimulating questions raised by the reviewers have motivated us to conduct the following additional numerical experiments (please see the attached pdf file), which add to the empirical evidence and the paper:

- We extended our empirical analysis of feature importance.
- We used real experiments to demonstrate the superiority of FDR control over other guarantees in the literature in terms of relevance to downstream tasks.

We also added more discussion in response to the reviewers’ important suggestions for clarification and further discussion, including

- unique challenges and contributions relative to conformal selection
- factors that may affect the power of our methods, and non-asymptotic power analysis
- potential extension to distribution shift, in-context learning, multi-task setting, and conversational settings with dependence
- potential extension to improve power and efficient data usage
- practical choice of the alignment score and predictor

Please see our point-by-point response to all questions raised by each reviewer’s comments below.

**Reference**

1. Angelopoulos, Anastasios N., et al. "Learn then test: Calibrating predictive algorithms to achieve risk control." arXiv preprint arXiv:2110.01052 (2021).
2. Angelopoulos, Anastasios N., et al. "Conformal risk control." arXiv preprint arXiv:2208.02814 (2022).
3. Quach, Victor, et al. "Conformal language modeling." arXiv preprint arXiv:2306.10193 (2023).
4. Mohri, Christopher, and Tatsunori Hashimoto. "Language models with conformal factuality guarantees." arXiv preprint arXiv:2402.10978 (2024).
5. Smit, Akshay, et al. "CheXbert: combining automatic labelers and expert annotations for accurate radiology report labeling using BERT." arXiv preprint arXiv:2004.09167 (2020).
6. Varshney, Neeraj, Swaroop Mishra, and Chitta Baral. "Investigating selective prediction approaches across several tasks in iid, ood, and adversarial settings." arXiv preprint arXiv:2203.00211 (2022).
7. Kumar, Bhawesh, et al. "Conformal prediction with large language models for multi-choice question answering." arXiv preprint arXiv:2305.18404 (2023).
8. Jin, Ying, and Emmanuel J. Candès. "Selection by prediction with conformal p-values." Journal of Machine Learning Research 24.244 (2023): 1-41.
9. Jin, Ying, and Emmanuel J. Candès. "Model-free selective inference under covariate shift via weighted conformal p-values." arXiv preprint arXiv:2307.09291 (2023).
10. Lin, Zhen, Shubhendu Trivedi, and Jimeng Sun. "Generating with confidence: Uncertainty quantification for black-box large language models." arXiv preprint arXiv:2305.19187 (2023).
11. Lorenz Kuhn, Yarin Gal, and Sebastian Farquhar. Semantic uncertainty: Linguistic invariances for uncertainty estimation in natural language generation. arXiv preprint arXiv:2302.09664, 2023.
12. Cherian, John J., Isaac Gibbs, and Emmanuel J. Candès. "Large language model validity via enhanced conformal prediction methods." arXiv preprint arXiv:2406.09714 (2024).

---

### Decision · Program_Chairs · 2024-09-25

**Decision:**

Accept (poster)

**Comment:**

This work addresses an important problem on the emerging topic of foundation models (and more particularly related to large-language models). Despite the reviewers raised several important points, and some may not be fully addressed in the rebuttal (e.g., novelty with respect to prior works), I believe that the proposed work has some merits, as mentioned by the reviewers. More specifically, the task of conformal prediction has not been explored in LLMs, despite its paramount importance. Furthermore, the experiments are extensive, which add empirical support to the mathematical formulation and theoretical justifications presented in this work. Thus, I recommend the acceptance of this submission.